# Functional fission of parvalbumin interneuron classes during fast network events

**Csaba Varga[1,2]\*, Mikko Oijala[1], Jonathan Lish[1], Gergely G Szabo[1], Marianne Bezaire[1], Ivan Marchionni[1], Peyman Golshani[3,4], Ivan Soltesz[1]**

[1]Department of Anatomy and Neurobiology, University of California, Irvine, Irvine, United States; [2]Szentágothai Research Center, MTA-PTE-NAP A-Entorhinal Microcircuits, Department of Physiology, University of Pécs, Pécs, Hungary; [3]Department of Neurology, David Geffen School of Medicine, University of California, Los Angeles, Los Angeles, United States; [4]West Los Angeles, VA Medical Center, Los Angeles, United States

**Abstract** Fast spiking, parvalbumin (PV) expressing hippocampal interneurons are classified into basket, axo-axonic (chandelier), and bistratified cells. These cell classes play key roles in regulating local circuit operations and rhythmogenesis by releasing GABA in precise temporal patterns onto distinct domains of principal cells. In this study, we show that each of the three major PV cell classes further splits into functionally distinct sub-classes during fast network events in vivo. During the slower (<10 Hz) theta oscillations, each cell class exhibited its own characteristic, relatively uniform firing behavior. However, during faster (>90 Hz) oscillations, within-class differences in PV interneuron discharges emerged, which segregated along specific features of dendritic structure or somatic location. Functional divergence of PV sub-classes during fast but not slow network oscillations effectively doubles the repertoire of spatio-temporal patterns of GABA release available for rapid circuit operations.

**\*For correspondence:** csaba.
varga@aok.pte.hu

**Competing interests:** The authors declare that no competing interests exist.

**Reviewing editor**: Frances K Skinner, University Health Network, and University of Toronto, Canada

## Introduction

Brain state-dependent network oscillations occurring at various frequencies provide multiscale temporal windows for the precise timing of neuronal discharges and the temporal binding of spatially distributed cell populations (*Singer, 1993*; *Buzsaki and Chrobak, 1995*). The spatio-temporal control of principal cell assemblies during endogenous brain rhythms is orchestrated by GABAergic interneurons (*Soltesz, 2006*; *Somogyi et al., 2014*). Versatility and precision in the GABAergic coordination of principal cell excitability arise from the diversity of interneurons that enables the selective timing of GABAergic inputs to specific spatial domains of principal cells. Within the CA1 region of the rodent hippocampus, 21 distinct interneuronal classes are currently recognized, each possessing unique input–output connectivity patterns, developmental profiles, and characteristic molecular and electrophysiological properties (*Klausberger and Somogyi, 2008*; *Bezaire and Soltesz, 2013*). The distinct features of interneuronal classes enable these cells to be selectively recruited to entrain principal cell populations into different oscillatory patterns of activity, including the theta, gamma, epsilon, and ripple waves that are associated with specific behaviors (*O'Keefe and Nadel, 1978*; *O'Keefe and Recce, 1993*; *Soltesz and Deschenes, 1993*; *Buzsaki and Chrobak, 1995*; *Lubenov and Siapas, 2009*; *Colgin and Moser, 2010*; *Maier et al., 2011*).

The parvalbumin expressing (PV) interneurons, in particular, have been identified to serve major mechanistic roles in a variety of network functions, including local circuit operations, learning and memory, rhythmogenesis, sensory processing, and critical period plasticity (*Pouille and Scanziani,*

**eLife digest** The brain continuously processes information from outside and inside the body to cope with the challenges of everyday life. As the brain carries out these processes, networks of neurons produce patterns of electrical activities called oscillations.

Fast-spiking PV cells are neurons that orchestrate the precise timing of these oscillations in a region of the brain called the hippocampus, which is important for the formation of memories. PV cells perform this role by releasing a chemical called GABA that suppresses electrical activity.

The hippocampus contains three distinct sub-classes of fast-spiking PV cells, but it is not clear how these different sub-classes collaborate to control the network oscillations in the hippocampus. Varga et al. have now explored this question by recording the electrical activity of PV cells in mice, while they were resting and also while they were running.

PV cells are involved in both fast and slow network oscillations. As had been found in previous experiments, Varga et al. found that the three different sub-classes of PV cells behaved similarly during slow network oscillations. During fast oscillations, however, the neurons within each sub-class displayed two distinct types of behavior, depending on their shape and location.

PV cells release GABA in patterns that depend on both space and time: the work of Varga et al. shows that the repertoire of patterns that can be employed by PV cells is about twice as big as was previously thought. Future studies are needed to explore the influence of this behavior on memory.

*2004*; *Buzsaki and Wang, 2012*; *Kuhlman et al., 2013*; *Siegle and Wilson, 2014*). Furthermore, PV cells have also been linked to a number of neurological and psychiatric disorders including epilepsy, schizophrenia, and autism (*Lewis et al., 2005*; *Ogiwara et al., 2007*; *Gibson et al., 2009*; *Armstrong and Soltesz, 2012*; *Verret et al., 2012*; *Trouche et al., 2013*). A key property of PV cells is speed; indeed, these interneurons can fire fast action potentials at high frequencies with little accommodation (*Kawaguchi, 1995*; *Buhl et al., 1996*), possess fast membrane time constants, and release GABA with high temporal precision following the arrival of presynaptic action potentials at the release sites, enabling PV cells to serve as the rapid signaling elements of interneuronal–principal cell networks (*Bartos et al., 2001*; *Jonas et al., 2004*; *Hu et al., 2010*). Importantly, PV cells are known to sharply segregate into three major classes that are differentiated based on the distinct post-synaptic domains that they innervate (*Klausberger and Somogyi, 2008*). Axo-axonic (chandelier) cells (AACs) make synaptic contacts exclusively on the axon initial segments of principal cells (*Somogyi, 1977*; *Somogyi et al., 1983*). In contrast, basket cells (BCs) innervate the somata and proximal dendrites of the target cells, whereas the bistratified (Bistrat) cells provide GABAergic inputs to the basal and apical dendritic domains (*Buhl et al., 1994*). In spite of the key functional importance of these specialized rapid signaling elements within the network, our understanding of the cellular sources of GABA from PV cells during fast network oscillations in vivo without anesthesia remains limited. For example, although AACs have received a lot of attention as playing major roles in the synchronization of principal cell populations (*Howard et al., 2005*; *Szabadics et al., 2006*; *Woodruff et al., 2010*), to date only a single AAC has been reported from the CA1 region under non-anesthetized in vivo conditions (*Viney et al., 2013*). Similarly, in spite of a series of recent studies examining anatomically and immunocytochemically identified (*Lapray et al., 2012*; *Varga et al., 2012*; *Viney et al., 2013*; *Katona et al., 2014*) or optogenetically 'tagged' interneurons in vivo without anesthesia (*Royer et al., 2012*; *Kvitsiani et al., 2013*), there are no data on the preferential phase-specific discharges of AACs and Bistrat cells during high-frequency (>90 Hz) oscillations in vivo in awake animals.

In this study, we report the brain state-specific action potential discharge patterns of AACs, BCs, and Bistrat cells during fast network oscillations in the CA1 region of the hippocampus from anesthesia-free head-fixed mice that were running or resting on a spherical treadmill (*Dombeck et al., 2007*; *Varga et al., 2012*). The cells were recorded using juxtacellular techniques (*Pinault, 1996*) that enabled the non-invasive recording (i.e., without disturbance of the intracellular milieu of the recorded neuron), high-fidelity labeling, and rigorous post-hoc identification of single cells. In addition to providing new insights into the precise patterns of discharges by AACs and Bistrat cells during brain state-dependent rhythms of the hippocampus, we report that during fast, but not slow, network oscillations each of the three major PV interneuron class splits into two functionally distinct sub-classes that

segregate according to characteristic morphological properties. These findings increase our understanding of the mechanisms of hippocampal oscillations and describe a unique form of functional fission of previously established PV cell classes that effectively doubles the available repertoire of spatio-temporal patterns of GABA release during rapid circuit operations.

## Results

### Identification of cells and oscillations

Fast spiking PV interneurons (total: n = 27; *Figure 1A*) were post-hoc identified primarily based on their axonal characteristics as BCs (n = 12; 7 were included in *Varga et al., 2012*), Bistrat cells (n = 8), or AACs (n = 7). A summary of the n = 27 PV cells, including the immunocytochemical tests and EM verification used for each cell, is presented in *Table 1*. Axons of the BCs were largely confined to the stratum pyramidale, where they formed synapses on the somata and proximal dendrites of PCs. Electron microscopic verification of somatic axonal targets of BCs was performed in n = 4 cells. In contrast, the dendritically projecting Bistrat cells had the majority of their axons in the stratum radiatum and some in the stratum oriens. Since the axons of Bistrat cells avoided the stratum pyramidale, the axons provided a feature of Bistrat cells that clearly distinguished them from the other two cell classes. The AACs projected to the stratum pyramidale, where they formed synapses exclusively on

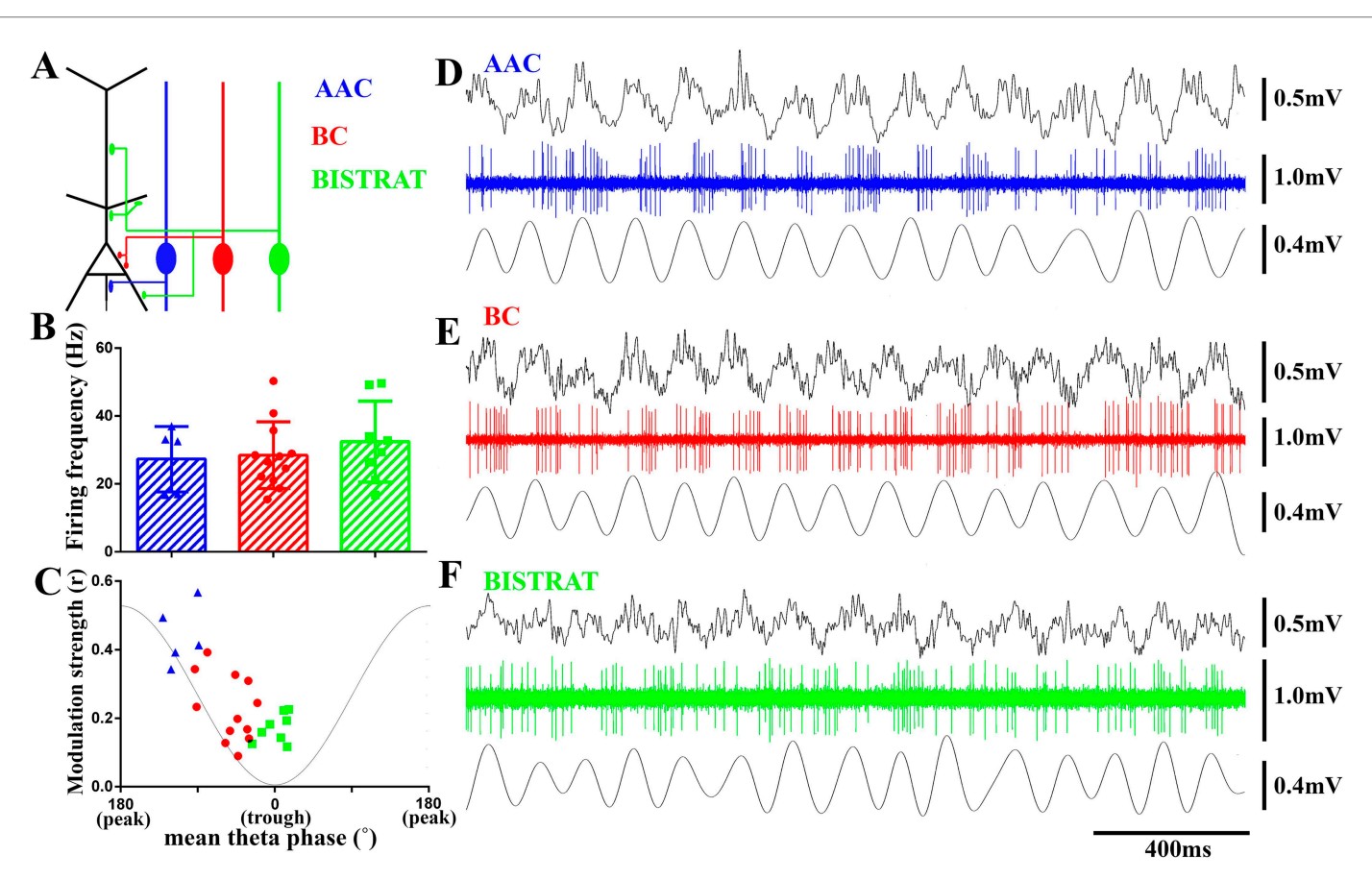

**Figure 1**. PV cells discharge at similar rates during running-associated theta oscillations but show between-class differences in theta modulation and preferred phases of firing. (**A**) The 3 classes of PV interneuron innervate distinct post-synaptic domains of CA1 PCs (black). AACs exclusively target the axon initial segments (AAC, blue); BCs innervate the perisomatic region (somata and proximal dendrites) (BC, red); Bistrat cells synapse on the apical and basal dendrites (Bistrat, green). (**B**) Firing frequencies during running are high and not significantly different between the 3 PV cell classes. (**C**) Mean preferred phase vs modulation strength of PV cell discharges. Blue triangles: AACs, red circles: BCs, green squares: Bistrat cells. A thin gray line representing a theta cycle is shown to illustrate the theta phases. (**D**–**F**) Example traces of LFP, spikes, and filtered theta oscillation of an AAC (**D**), BC (**E**), and Bistrat cell (**F**). Scales: LFPs = 0.5 mV; single cell unit firing = 1 mV; theta = 0.4 mV.

**Table 1.** List of all n = 27 in vivo juxtacellularly filled PV interneurons and their identification based on immunocytochemistry and electron microscopy (EM)

| Cell ID# | Cell type | Parvalbumin | Somatostatin | mGluR1a | SATB1 | Bouton targets | |
|---|---|---|---|---|---|---|---|
| | | | | | | AnkyrinG verification | EM verification |
| 123011c3 | C-BC | + | n.t. | n.t. | n.t. | n.t. | Soma/proximal dendrite |
| cvi30 | C-BC | + | n.t. | n.t. | n.t. | n.t. | Soma/proximal dendrite |
| cvi33 | C-BC | + | n.t. | n.t. | n.t. | n.t. | n.t. |
| cvi35 | C-BC | + | n.t. | n.t. | n.t. | n.t. | n.t. |
| cvi55 | C-BC | n.t. | n.t. | n.t. | n.t. | n.t. | Soma/proximal dendrite |
| cvi65 | C-BC | + | n.t. | n.t. | n.t. | n.t. | n.t. |
| cvi75 | C-BC | + | n.t. | n.t. | n.t. | n.t. | n.t. |
| cvi151 | C-BC | n.t. | n.t. | n.t. | n.t. | n.t. | n.t. |
| cvi251 | C-BC | n.t. | n.t. | n.t. | n.t. | n.t. | n.t. |
| cvi240b | H-BC | + | n.t. | n.t. | n.t. | n.t. | Soma/proximal dendrite |
| 042911c5 | H-BC | + | n.t. | n.t. | n.t. | n.t. | n.t. |
| gs0920 | H-BC | + | n.t. | n.t. | n.t. | n.t. | n.t. |
| gs012913 | O-Bistrat | + | + | n.t. | n.t. | n.t. | n.t. |
| gs022713 | O-Bistrat | + | + | n.t. | n.t. | n.t. | n.t. |
| cvi017 | O-Bistrat | n.t. | + | n.t. | n.t. | n.t. | n.t. |
| cvi255 | O-Bistrat | − | + | n.t. | n.t. | n.t. | n.t. |
| cvi312 | O-Bistrat | n.t. | + | − | n.t. | n.t. | n.t. |
| cvi270 | C-Bistrat | + | + | n.t. | n.t. | n.t. | n.t. |
| imi069 | C-Bistrat | n.t. | + | n.t. | n.t. | n.t. | n.t. |
| cvi190 | C-Bistrat | − | + | n.t. | n.t. | n.t. | n.t. |
| 090311c3 | E-AAC | + | n.t. | n.t. | n.t. | n.t. | AIS |
| cvi059 | E-AAC | n.t. | n.t. | n.t. | n.t. | n.t. | AIS |
| cvi153 | C-AAC | n.t. | n.t. | n.t. | n.t. | n.t. | AIS |
| cvi258 | C-AAC | + | n.t. | n.t. | n.t. | n.t. | AIS |
| cvi315 | C-AAC | n.t. | n.t. | n.t. | − | n.t. | n.t. |
| imi075 | C-AAC | + | n.t. | n.t. | n.t. | AIS | n.t |
| 07082014cs6 | C-AAC | n.t. | n.t. | n.t. | n.t. | AIS | AIS |

Abbreviations: n.t.: not tested; AIS: axon initial segment.

the axon initial segments. The targets of n = 5 AACs were identified at the EM level (number of boutons examined: 11, 10, 10, 8, 4, all synapsing on axon initial segments). Out of the two AACs not tested at the EM level, one AAC was tested for ankyrinG, a marker for axon initial segments (*Jenkins and Bennett, 2001*). The axon terminals of the tested AACs were found to be in close juxtaposition with ankyrinG immunopositive profiles (not shown). The remaining AAC was tested for the expression of special AT-rich sequence binding protein 1 (SATB1), because SATB1 is expressed in BCs and Bistrat cells but not in AACs (*Viney et al., 2013*). As expected, the AAC was found to be immunonegative for SATB1 (not shown).

In addition, PV or somatostatin (SOM) immunoreactivity was also tested. Specifically, PV immunopositivity was confirmed in 8 out of the 8 tested BCs, 3 out of the 5 tested Bistrat cells, and 3 out of the 3 tested AACs. SOM immunopositivity was verified in all 8 out of the 8 tested Bistrat cells, which is important since SOM expression distinguishes Bistrat cells from BCs and AACs (*Klausberger and Somogyi, 2008*). Note that Bistrat cells have been observed to exhibit generally weaker PV immunoreactivity compared to BCs (*Ferraguti et al., 2004*), in agreement with recent reports indicating often low PV immunopositivity in cells with concurrently active PV and SOM promoters (*Fenno et al., 2014*).

Because the nomenclature of the various fast oscillations is not always consistent across various studies, it is important to clearly state how we used the various terms relating to fast rhythms. Specifically, we examined PV cell discharges during four major oscillations that appeared either when the animal was running or resting, and we used these two distinct behavioral states that could be objectively distinguished in our head-fixed paradigm as an important factor in classifying oscillations. Accordingly, theta (5 Hz–10 Hz), gamma (25 Hz–90 Hz), and epsilon (90 Hz–130 Hz) oscillations were studied during running, whereas ripples (90 Hz–200 Hz) occurred during rest.

Regarding the running-associated high-frequency oscillations, a unique frequency band has been recently distinguished called epsilon (*Freeman, 2007*; *Belluscio et al., 2012*; *Bieri et al., 2014*; *Schomburg et al., 2014*). Epsilon is distinct from other high frequency oscillations in terms of generation mechanisms and location. Specifically, epsilon oscillations reflect network events generated locally in CA1 (*Schomburg et al., 2014*), and most pyramidal cells (PCs) show strong phase locking to the trough of the epsilon oscillations, indicating large synchrony of PC discharges during these events (*Belluscio et al., 2012*; *Schomburg et al., 2014*). Another important point regarding the nomenclature of fast rhythms is that our running-associated epsilon oscillations are similar, but not identical, to the high gamma oscillations recorded during theta in other studies (*Csicsvari et al., 1999a*; *Canolty et al., 2006*; *Colgin et al., 2009*), because these latter studies employed a generally wider frequency band for detecting high gamma oscillations compared to our filter setting for epsilon.

Regarding the fast rhythms that occur during rest, the 90 Hz–200 Hz fast oscillations are sometimes differentiated into 'high' (or 'fast') gamma (90–140 Hz) and ripple (140–200 Hz) oscillations (*Csicsvari et al., 1999b*). However, because the non-theta-associated 90 Hz–140 Hz and 140 Hz–200 Hz fast oscillations during rest can appear under similar behavioral conditions, share similar generation mechanism, and differ mostly in the strength of incoming CA3 input (*Sullivan et al., 2011*), for the purposes of the current study we considered them together under the collective term 'ripples' (90 Hz–200 Hz). This approach avoids the sorting of events according to a sharp 140 Hz boundary into 90–140 Hz or 140 Hz–200 Hz oscillations and still allows the comparative study of the firing of interneurons during transient events that fall towards the lower- vs higher-end of the continuum of 90 Hz–200 Hz oscillations (see below).

Most cells in our sample could be analyzed for running-associated theta, gamma, epsilon, and rest-associated ripple oscillations, with the exception of n = 2 AACs (cells imi075 and 07082014cs6 in *Table 1*) that yielded data only for ripples, but not for theta, gamma, or epsilon oscillations due to the presence of only brief periods of running and theta waves.

## PV cells show class-dependent differences in modulation strengths and preferred phases of firing during theta oscillations

We first examined the action potential firing of BCs, Bistrat cells, and AACs during running-associated theta oscillations. The three distinct cell classes discharged at comparable frequencies during theta (*Figure 1B,D–F*; AAC: 27.1 ± 9.1 Hz, n = 5; BC: 28.64 ± 9.4 Hz, n = 12; Bistrat: 34.03 ± 11.9 Hz, n = 8; p = 0.84, one way ANOVA). While every cell showed theta modulation of its firing (p < 0.001 for all cells, Rayleigh test), the strength of theta modulation differed between the three cell classes. Specifically, AACs showed the most prominent theta modulation, followed by the intermediate level of modulation of the BCs, whereas Bistrat cells were only weakly modulated (*Figure 1D–F*). Statistical analysis of the strength of theta modulation of action potential discharges revealed significant differences between the three cell classes (modulation strength, from the highest to low: AACs: r = 0.44 ± 0.1; BCs: r = 0.25 ± 0.1; Bistrat cells: r = 0.17 ± 0.05; p < 0.01; Kruskal–Wallis test; *Figure 1C*).

The three cell classes also displayed preferential firing at distinct phases of the theta waves. The AACs preferred the middle of the descending phase (251° ± 17°; 0° is considered to be the trough, 180° the peak of the individual oscillatory cycle). Thus, AACs did not preferentially fire close to the theta peak during running, as they were reported to do so under anesthesia (*Klausberger et al., 2003*). The preferential discharges of the AACs during the theta cycle were followed by the BCs that fired on average during the late descending phase (310° ± 23°), while Bistrat cells discharged near the trough (0° ± 17°), around the time that the mean preferential discharges of most PCs take place (*Mizuseki et al., 2011*). These data suggest that the different PV interneuron classes deliver GABA in a sequential manner along the long (axon initial segment–somato-dendritic) axis of the CA1 PCs, and that the bulk of this spatio-temporally organized inhibition sweeps through the PC populations from the axon initial segment to the dendrites within about 40 ms during running-associated theta (calculated for an average, 8 Hz theta rhythm).

The BC, AAC, and Bistrat cell groups occupied different parts of the phase vs modulation strength plot (*Figure 1C*), with the earliest firing cells (the AACs) also being the strongest modulated, while Bistrat cells fired last during the theta cycle and with the least amount of theta modulation. As illustrated in *Figure 1C* for the AACs, BCs, and Bistrat cells, there was a high degree of within-class uniformity in how members of these three cell groups discharged during theta oscillations. On the other hand, it should also be noted that even though the AACs, BCs, and Bistrat cells comprised statistically different groups in terms of their average strength of theta modulation and phase of preferential firing, the cell clusters were close to each other in the phase vs modulation strength plot in *Figure 1C*. Therefore, extracellularly recorded fast spiking single units (e.g., recorded with tetrodes or silicon probes) without direct anatomical identification cannot be unambiguously classified as belonging to one of these three cell classes based on the theta-related firing alone, even if the unit is identified as expressing PV (e.g., with optogenetic tagging; [*Royer et al., 2012*]).

## Gamma oscillation-related PV cell firing

Gamma oscillations are thought to play important roles in information transfer between brain areas (*Engel and Singer, 2001*; *Womelsdorf et al., 2007*; *Sirota et al., 2008*; *Atallah and Scanziani, 2009*), with the synchronous gamma-frequency firing in groups of spatially distinct cells hypothesized to contribute to the 'binding' of information in downstream neuronal groups (*Engel and Singer, 2001*). While the precise roles of gamma rhythm in information processing are not fully understood, there is an overall agreement that perisomatic inhibition is mechanistically important in the generation of gamma-frequency oscillations (*Buzsaki and Wang, 2012*). Gamma oscillations appear during the falling phase of the running-associated theta rhythm (see peak gamma activity centered around 60 Hz in *Figure 2A*). Interestingly, although fast spiking PV cells are important in gamma oscillations (*Buzsaki and Wang, 2012*), not all PV cells in our recordings fired in a gamma-modulated manner (number of cells significantly modulated: BCs: 9/12; Bistrat cells: 6/8; AACs: 4/5). Furthermore, the modulation strength was only moderate (in agreement with *Tukker et al., 2007*; *Varga et al., 2012*), and there was no difference between the perisomatically (BCs, AACs) vs dendritically (Bistrat cells) projecting PV interneurons in terms of the strength of modulation (BC: $r = 0.18 \pm 0.07$; Bistrat cells: $r = 0.18 \pm 0.05$; AAC: $r = 0.17 \pm 0.08$; $p = 0.79$; Kruskal–Wallis test). BCs preferentially fired during the middle ascending phase of the individual gamma waves ($110 \pm 18°$), significantly earlier than the other two cell classes ($p < 0.01$; Watson–Williams test) that discharged on average during the late ascending phase (Bistrat cells: $148 \pm 23°$) or close to the peak of the gamma oscillations (AACs: $181 \pm 15.4°$). These data indicate that the majority of cells in each PV class was moderately modulated by gamma-frequency oscillations, and that the BCs preferentially fired earlier during the individual gamma waves compared to either the Bistrat cells or the AACs.

## PV interneuron discharges during high-frequency oscillations associated with running (theta-embedded epsilon waves)

Theta oscillations themselves are relatively slow, but high-frequency (>90 Hz) oscillations are known to be also present during the theta rhythm (*Canolty et al., 2006*; *Colgin et al., 2009*; *Belluscio et al., 2012*). The epsilon oscillations occurred during the second half of the falling phase of the running-associated theta waves (asterisk in *Figure 2A*). From the cell class-specific theta-related firing of PV interneurons described above and the relative timing of epsilon oscillations during theta, we predicted that AACs were unlikely to be strongly modulated by these running-associated high-frequency oscillations. Specifically, as illustrated in *Figure 2B* (re-plotted from the data in *Figure 1C* to emphasize the distribution of firing probabilities during theta), the peak firing of AACs took place slightly before the peak of the epsilon oscillations (asterisk in *Figure 2A*). Indeed, AACs were either not epsilon modulated (n = 3) or they exhibited only weak epsilon modulation (n = 2; blue triangles in *Figure 2C*; $r = 0.19$ and 0.12; y-axis in *Figure 2C*).

In contrast to AACs, the firing probabilities of every BC and Bistrat cell showed significant epsilon modulation (*Figure 2C*). On average, the modulation strength of the BCs was higher than that of the Bistrat cells (BC: $r = 0.38 \pm 0.09$, Bistrat: $r = 0.26 \pm 0.07$; $p < 0.01$, Mann–Whitney test). These data indicate a more precise epsilon-related firing activity of BCs compared to Bistrat cells (and AACs), which likely contributes to the regulation of PC discharges during epsilon epochs (*Belluscio et al., 2012*; *Schomburg et al., 2012*).

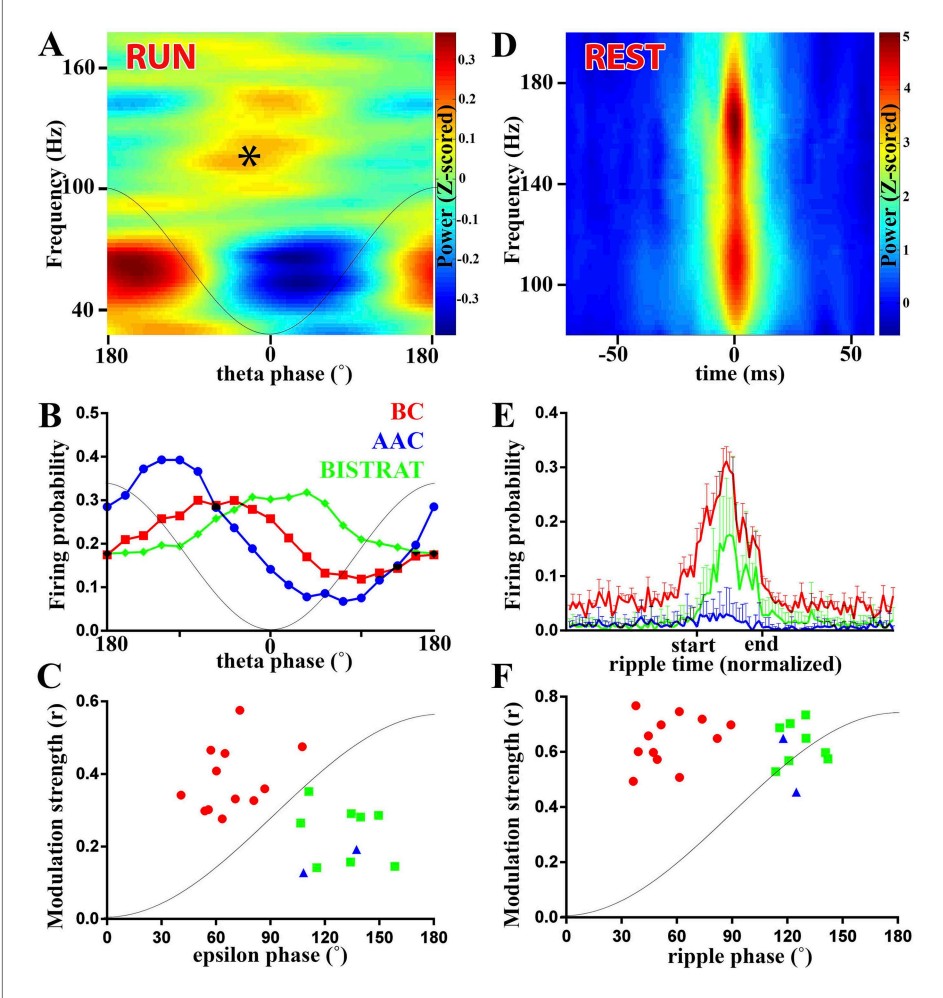

**Figure 2**. PV cell discharges during high-frequency (>90 Hz) oscillations associated with running or resting. (**A–C**) Running-associated, theta-nested epsilon oscillations. (**A**) Averaged time–frequency plot triggered by theta troughs during running. A thin gray line representing a theta cycle is overlaid on the plots in panels **A** and **B** to illustrate the theta phases. Asterisk indicates the epsilon oscillations occurring during the late descending phase of theta. (**B**) Average firing probabilities of the 3 PV cell classes during theta oscillations. Note that the peak firing probability of BCs and Bistrat cells overlaps with the occurrence of epsilon oscillatory epochs (asterisk in panel **A**). (**C**) Preferred phase of firing and strength of modulation of discharges during epsilon oscillations (individual cells). A thin gray line representing half an epsilon cycle is overlaid on the plot to illustrate the epsilon phases. (**D–F**) Rest-associated ripples. (**D**) Time–frequency plot of detected ripples (90–200 Hz) during resting. (**E**) The mean firing probabilities of PV cell classes before, during, and after ripples (error bars: S.D.). (**F**) Preferred phase of firing and strength of modulation of discharges during ripple oscillations (individual cells). A thin gray line representing half a ripple cycle is overlaid on the plot to illustrate the ripple phases. Blue triangles: AACs, red circles: BCs, green squares: Bistrat cells.

PV interneurons were not only modulated by epsilon, but they also showed differential phase preferential firing with respect to the epsilon waves (x-axis in *Figure 2C*; as with the theta waves, 0° is the trough, 180° is the peak of the individual epsilon oscillatory cycle). The BCs on average fired significantly earlier during the epsilon waves compared to the Bistrat cells (BCs: 66 ± 17°, n = 12; Bistrat cells: 126 ± 14°, n = 8; p < 0.001; Watson–Williams test; 1–1.5 ms delay); the 2 significantly epsilon-modulated AACs preferentially fired at phases similar to Bistrat cells (137° and 108°; *Figure 2C*).

Taken together, these data show that the firing of PV interneurons segregates according to the three established major PV cell classes during epsilon oscillations that take place when the animal is running.

# Inter-class differences in PV interneuron discharges during high-frequency oscillations during rest

High-frequency (>90 Hz) oscillations are not confined to running-associated epsilon waves but are well known to be present also during rest. In fact, the rest-associated high frequency oscillations have at least an order of magnitude higher increase in power than the running-associated epsilon oscillations (compare the heat-colored, Z-scored power above 90 Hz in *Figure 2A* vs *Figure 2D*). The rest-associated high-frequency oscillations typically appear in the 90 Hz–200 Hz frequency range (see *Figure 2D*) (*Csicsvari et al., 1999a*; *Klausberger et al., 2003*) and are collectively referred to as ripples in this paper (see above). Ripples, which are known to be associated with sharp-waves in the CA1 area (*O'Keefe and Nadel, 1978*; *Buzsaki et al., 1983*), represent one of the fastest (up to 200 Hz, lasting approximately 40 ms–150 ms; for an example, see *Figure 3*) and most synchronized episodic electrographic events in the normal brain (*Buzsaki, 1986*) and are crucial for memory consolidation by replaying the encoded ensemble firing patterns in a time-compressed manner (*Wilson and McNaughton, 1994*; *Foster and Wilson, 2006*).

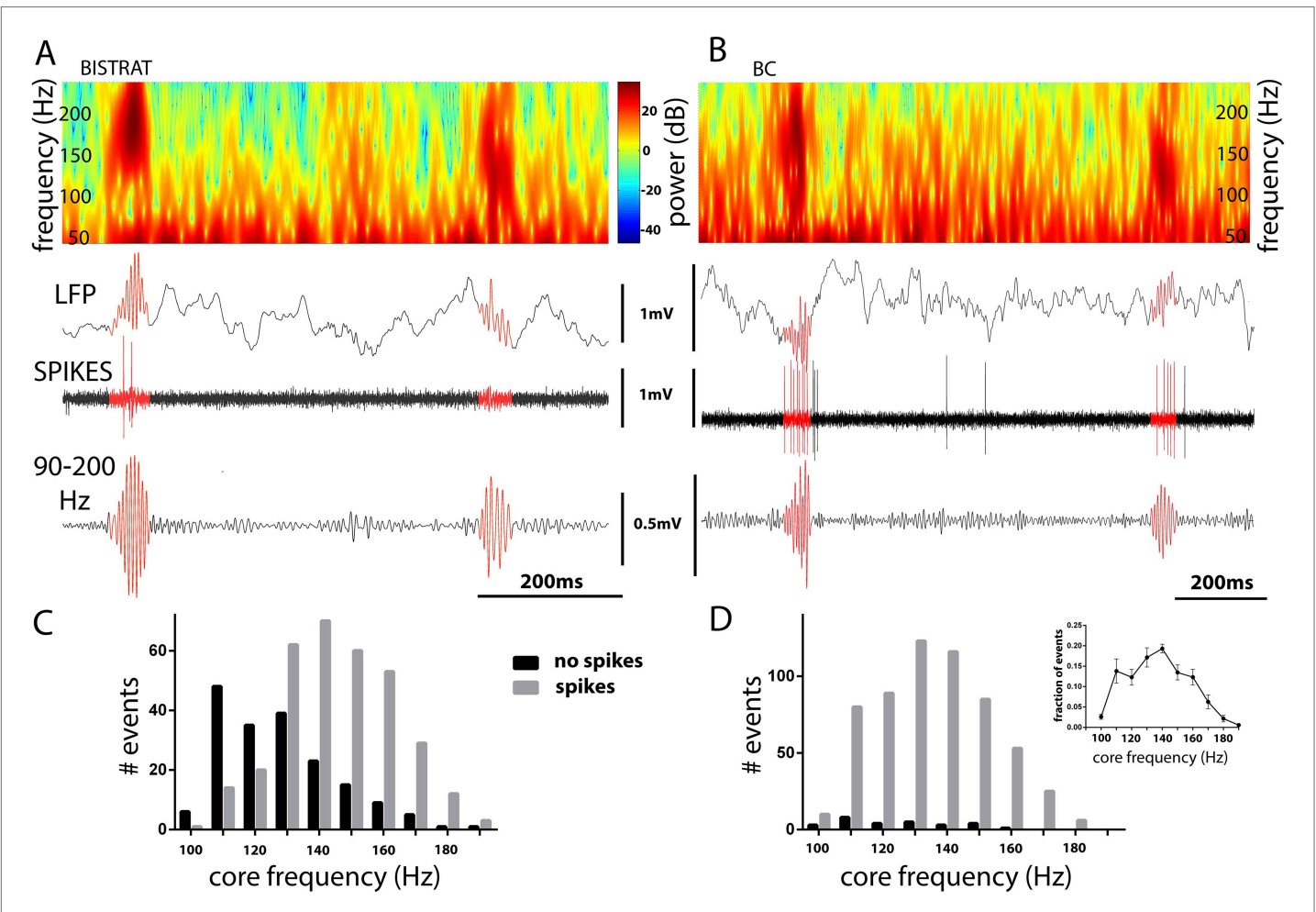

**Figure 3**. Bistrat cells, in contrast to BCs, differentially fire during faster and slower ripples. (**A** and **B**) Time–frequency representation (top) of example traces (LFP) with two high-frequency epochs (highlighted in red). Note that spikes of the recorded Bistrat cell (**A**) occurred only during the first, higher frequency oscillatory event, but remained silent during the second, lower frequency one. However, the BC (**B**) was highly active during both the higher and lower frequency epoch. (**C** and **D**) Frequency distribution of ripple events with (gray bars) or without (black bars) Bistrat cell (**C**) or BC (**D**) spiking (10 Hz binned). Data are pooled from all recorded Bistrat cells (n = 8, **C**) and BCs (n = 12, **D**). Note that Bistrat cells selectively fired during the higher frequency events but BCs discharged during virtually all ripple events. Inset in (**D**) illustrates that the high-frequency events showed a continuum of core frequencies (n = 16 animals; note that mean and standard error are plotted in the inset to facilitate comparison with *Figure 1B* in *Sullivan et al., 2011*).

How do the three classes of fast spiking interneurons discharge during ripples in non-anesthetized animals? Although the ripple-related firing rates of BCs (*Lapray et al., 2012*), Bistrat cells (*Katona et al., 2014*), and a single AAC (*Viney et al., 2013*) have been reported in recent studies from the CA1 region of non-anesthetized animals, we re-examined the firing of our PV interneurons during ripple events in order to gain a better understanding of the relative firing rates, strength of modulation, and the phase-preferential firing of these three cell classes. In agreement with previous reports (*Lapray et al., 2012*; *Katona et al., 2014*), BCs discharged at a high rate (77 ± 28 Hz) during ripple events, dramatically increasing their discharge probability (*Figure 2E*). Bistrat cells also significantly increased their discharge probabilities during ripples, but their firing frequency was about half of that of the BCs during the ripple events (36 ± 16 Hz, p < 0.01 with respect to BCs, Mann–Whitney; *Figure 2E*). In contrast to BCs and Bistrat cells, the AACs on average, as a group (n = 7), showed no significant changes in firing rates during ripple events (blue line in *Figure 2E*). Further analysis showed that the AACs fired the majority (68.3 ± 17.3%, n = 7) of their ripple-related spikes during the beginning of the ripple events (before the ripple reached its maximal amplitude), in broad agreement with the discharge dynamics noted by *Klausberger et al., 2003*.

The ripple-related modulation strengths of the firing of BCs and Bistrat cells were high and not different from each other (BCs: r = 0.63 ± 0.12; Bistrat: r = 0.62 ± 0.08). Therefore, Bistrat cells were weakly modulated during the running-associated epsilon waves but strongly modulated during the rest-associated ripples (*Figure 2C,F*). In terms of phase-preferential firing during ripples, the BCs on average discharged significantly earlier than the Bistrat cells (55 ± 20° vs 126 ± 10°, p < 0.001, Watson–Williams test), with a sharp division between the two clusters representing the BCs (red circles) and the Bistrat cells (green squares) in the phase vs modulation strength plot in *Figure 2F*.

Because BCs and Bistrat cells discharged at different rates during ripples (*Figure 2E*), we further examined what factors may influence the ripple-related firing of these two PV cell classes. Our analysis revealed that BCs discharged on virtually every ripple event (in agreement with [*Varga et al., 2012*]), whereas Bistrat cells remained silent (*Figure 3A,B*) on about a third of the ripple episodes (percentage of ripple events when the interneurons fired: BCs: 95 ± 6%; Bistrat cells: 68 ± 13%; p < 0.01, Mann–Whitney). Furthermore, the data showed that whether or not a Bistrat cell fired during a ripple correlated with the oscillation frequency of the individual ripple events (i.e., with the intra-ripple or 'core' frequency of the LFP oscillations; these are highlighted in red in the example traces in *Figure 3A,B*). Namely, as illustrated in *Figure 3C,D*, the core frequency of the ripples when Bistrat cells discharged was significantly higher than the core frequency of those ripples during which the Bistrat cells did not discharge (mean core frequency of ripple events without Bistrat cell spikes: 122.5 ± 18 Hz, n = 191 events; with spikes: 140 ± 17 Hz, 347 events; n = 8 cells, p < 0.0001, Mann–Whitney test). Therefore, these results indicate that, in sharp contrast to BCs (*Figure 3B,D*), Bistrat cells (*Figure 3A,C*) differentiate between lower vs higher frequency ripple events (note that, as illustrated in the inset in *Figure 3D*, rest-associated high-frequency events in our recordings showed no evidence of a bimodal distribution with a minimum at 140 Hz, the frequency boundary used to separate high gamma and ripple events in sleeping or awake immobile rats [see *Figure 1B* in *Sullivan et al., 2011*]).

Taken together, these data show that both the epsilon- and the ripple-associated discharges of PV interneurons display characteristic properties that segregate along previously established class lines. Furthermore, these results demonstrate that the phase-specific discharges of BCs characteristically preceded Bistrat cell firing during fast oscillations (>90 Hz), regardless of the behavioral state of the animal (epsilon during running, ripples during rest).

## Ripple-related intra-class differences among Bistrat cells segregate with dendritic structure

The Bistrat cells have been reported to occur in at least two distinct forms. One sub-class has extensive dendrites both in the oriens and the radiatum layers, with the dendrites in the stratum radiatum extending up to the border with the lacunosum-moleculare layer (*Klausberger et al., 2004*; *Katona et al., 2014*), which presumably allows these cells to receive inputs both from the local CA1 PCs and the Schaffer collateral/commissural inputs from the CA3 PCs (note that axons from CA3 PCs are present in the CA1 stratum oriens as well [*Sik et al., 1993*; *Wittner et al., 2006*]). We will refer to these cells as classical or C-Bistrat cells. In contrast, other Bistrat cells have their entire dendritic tree confined to the stratum oriens with no dendrites in the stratum radiatum (*Maccaferri et al., 2000*); these cells are referred to as oriens or O-Bistrat cells. From our n = 8 Bistrat cells, 3 were C-Bistrat

(*Figure 4A,C and 5*) were O-Bistrat cells (*Figure 4B,C*). Note that both sub-classes provided dense axonal projections to the radiatum, with minor innervation of the oriens layers (*Figure 4A,B*).

In spite of the marked differences in dendritic trees in the radiatum layer, there have been no functional differences reported in the literature between the C- and O-Bistrat cells, possibly because no previous study had both sub-classes sampled in vivo under the same conditions. Indeed, there was no difference in the running-associated theta-related firing of C- and O-Bistrat cells in our sample either (firing frequency during theta: O-Bistrat: 34.6 ± 14.7 Hz; C-Bistrat: 27.2 ± 6.5 Hz; p = 0.67, Mann–Whitney; *Figure 4D*; theta phase preference: O-Bistrat: 2.6 ± 14.6°; C-Bistrat: 357.6 ± 12.3°; strength of modulation of firing during theta, r: O-Bistrat: 0.17 ± 0.06; C-Bistrat: 0.19 ± 0.04; p = 0.7, Mann–Whitney). In contrast, during rest-associated ripples, O-Bistrat cells fired significantly faster than C-Bistrat cells (O-Bistrat: 44.2 ± 11.0 Hz; C-Bistrat: 19.8 ± 3.7 Hz; p = 0.035, Mann–Whitney; *Figure 4E*), with no differences in other aspects of ripple-related firing (phase preference: O-Bistrat: 122.41 ± 6.63°; C-Bistrat: 134.07 ± 10.35°; strength of modulation during ripples, r: O-Bistrat: 0.65 ± 0.09; C-Bistrat: 0.57 ± 0.03; p = 0.25, Mann–Whitney). The different firing rates of the O-Bistrat and C-Bistrat cell subclasses during ripples demonstrate the emergence of intra-class functional differences among Bistrat cells during fast network oscillations. In addition, the fact that O-Bistrat cells had especially high firing frequencies during ripples suggests that dendrites in the radiatum are not required for elevated levels of discharges in these cells during ripple events.

## Intra-class differences between AACs during fast network oscillations

Next, we asked the question if there were other functional distinctions beyond the ripple-associated firing of O- vs C-Bistrat cells during fast network oscillations between cells that belong to the same PV cell class. One potential clue was our observation that, as mentioned above, 2 out of the 5 AACs for whom theta, gamma, and epsilon-related firing could be analyzed showed significant modulation of

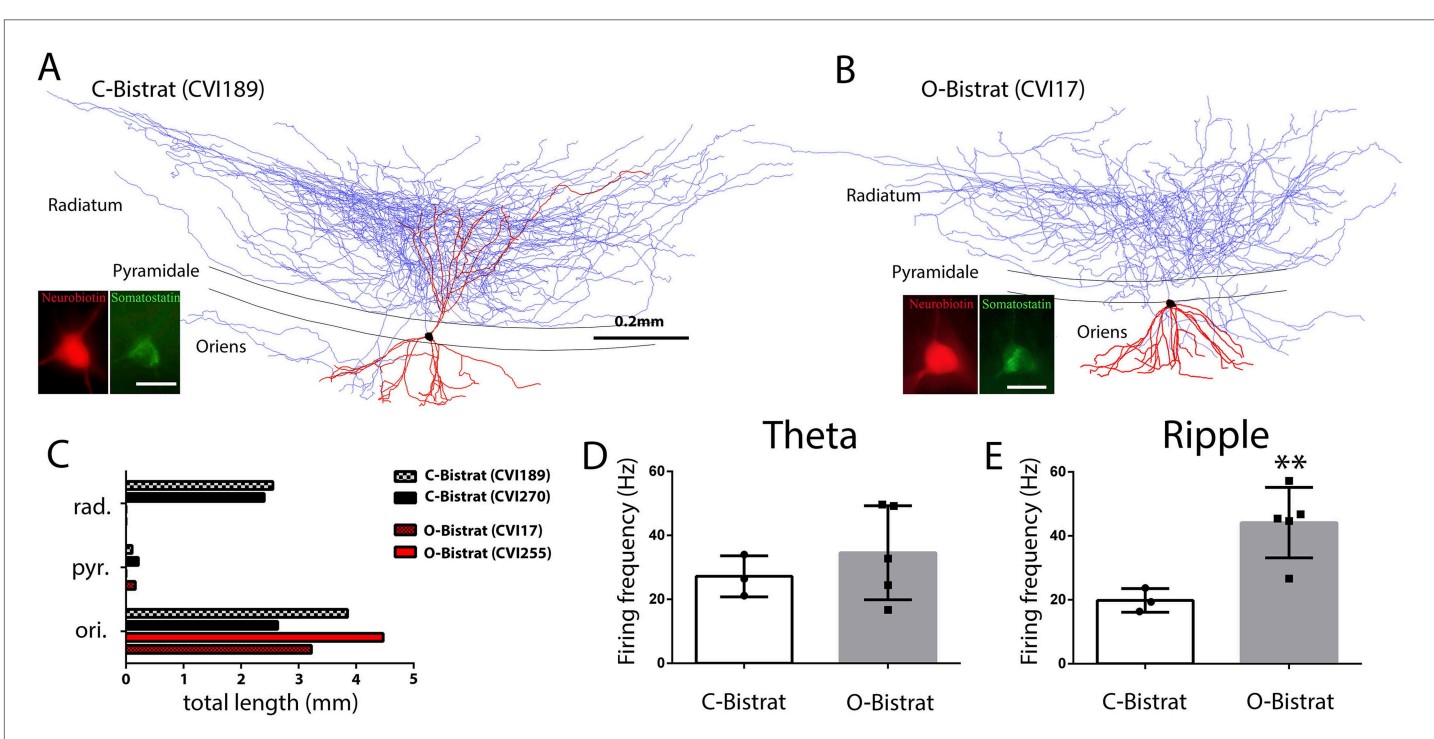

**Figure 4**. Morphologically distinct Bistrat cell sub-classes fire differentially during sharp-wave associated ripple events. (**A** and **B**) Fully reconstructed (axons: blue; dendrites: red; soma: black) representative cells of the two Bistrat cell sub-classes. (**C**) Total dendritic lengths measured in stratum radiatum/pyramidale/oriens of individual cells, with each cell differentially color coded. The O-Bistrat cells (**B**) have no dendrites in the radiatum, the C-Bistrat cells (**A**) have dendrites in stratum radiatum and oriens as well. (**D**) The firing frequencies of the O-bistrat and C-bistrat cells did not show significant differences during running-associated theta oscillations. (**E**) The O-Bistrat cells discharged at significantly higher rates than the C-Bistrat cells during ripples.

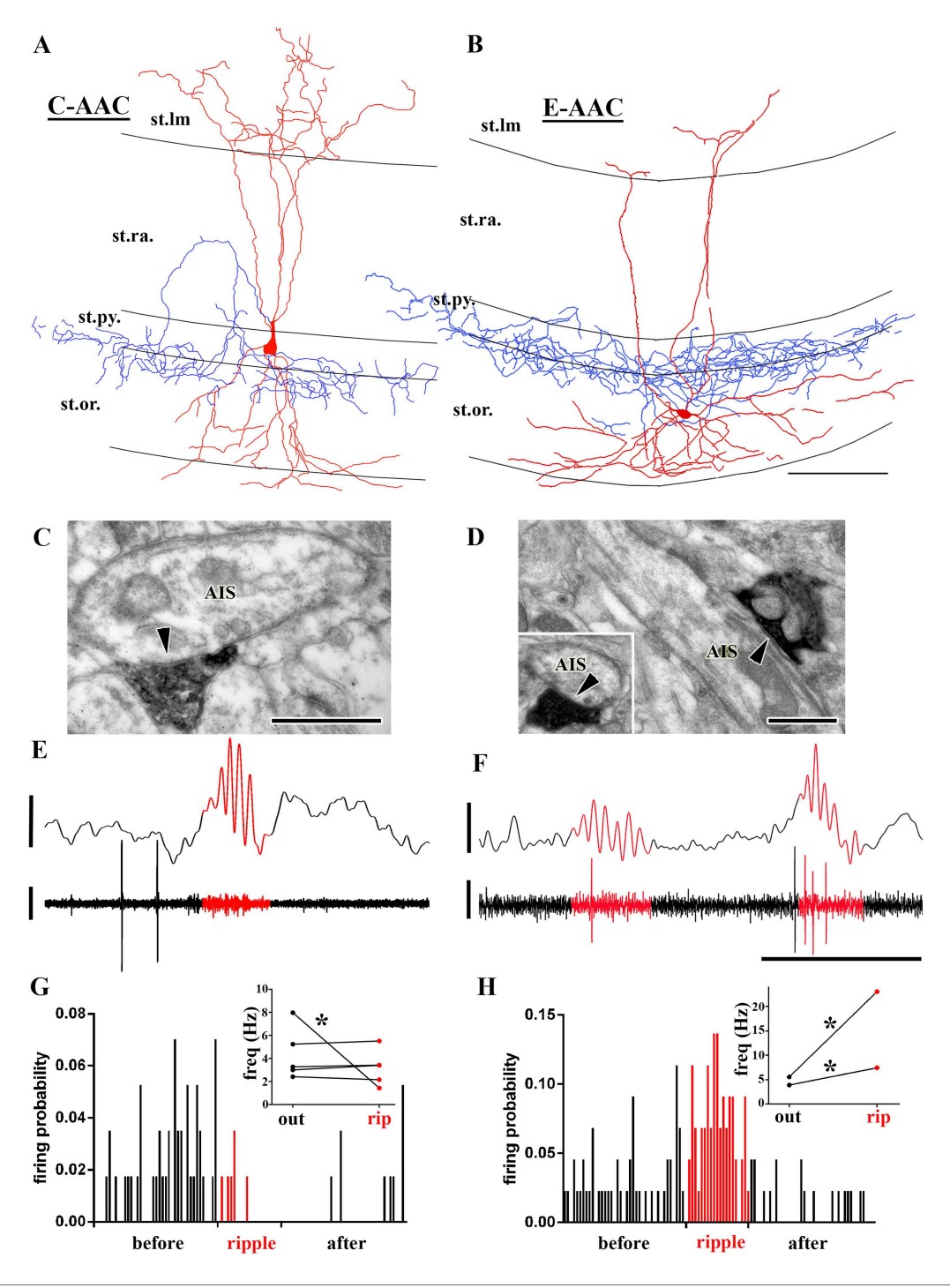

**Figure 5**. Sub-populations of AACs show distinct action potential discharge probabilities during ripples.
(**A** and **B**) Full reconstruction of dendritic arborizations and cell bodies of two representative AAC cells. Note that
the cell body in (**A**) is located within stratum pyramidale (C-AAC), but the cell in (**B**) is in the stratum oriens (E-AAC;
scale: 100 μm). (**C** and **D**) Electron microscopic verification of axon initial segments (AISs) as the exclusive post-synaptic
targets of the axons of AACs in (**A**) and (**B**), respectively. The bouton in (**C**) innervates two adjacent AISs. Panel (**D**)
also illustrates that targets were verified on longitudinally and transversely (inset) sectioned AISs (arrowheads in **C**,
**D**: synaptic clefts, scales: 500 nm). (**E** and **F**) Example LFPs (upper traces) and the action potentials of cells shown in
(**A**) and (**B**), respectively. Ripples are highlighted with red. Note the different firing activity of cell (**A**) and (**B**) during

*Figure 5. Continued on next page*

*Figure 5. Continued*

the events. Voltage scales: 0.5 mV; time scale: 100 ms. (**G** and **H**) Firing probabilities of cells in (**A**) and (**B**), respectively, as a function of normalized time before, during, and after ripples. Insets in (**G** and **H**): average firing frequencies of individual cells outside ('out'; black dots) and inside ('rip'; red dots) of ripple events. C-AACs (G, n = 5) did not change significantly their firing frequency during ripples; however, E-AACs (n = 2) showed elevated activity during events (statistical analysis as in *Varga et al. (2012)*, based on randomization of spike times; see 'Materials and methods'). Significance is indicated with asterisks (p < 0.05).

action potential discharges during running-associated epsilon waves (*Figure 2C*), indicating a certain amount of intra-class heterogeneity among AACs as well.

Did the AACs that showed significant epsilon-related modulation of their firing display any distinct anatomical features? Indeed, we noted that the somata of five of our AACs were situated within the PC layer or at the PC layer and stratum oriens border (*Figure 5A,C*; we refer to these cells as Classical or C-AACs), whereas the somata of the two AACs that showed epsilon-modulation were located outside of the PC layer (*Figure 5B,D*; we refer to these cells as External or E-AACs. Such E-AACs have been also noted before [*Ganter et al., 2004*; *Forro et al., 2013*]).

In addition, the C-AACs and E-AACs also differed in their firing during ripples (number of AACs that could be analyzed for ripples: n = 7). As illustrated in *Figure 5E,G*, C-AACs either did not alter (4 out of 5) or decrease (1 out of 5) their firing rates during ripples (inset in *Figure 5G*). In sharp contrast to the C-AACs, the two E-AACs significantly increased their discharge rates during ripples (*Figure 5F,H*; from 5.6 Hz–23 Hz, and from 3.8 Hz–7.5 Hz; p < 0.001 for each cell). Furthermore, the strength of modulation of the ripple-related firing by the E-AACs was strong (r = 0.69 and r = 0.45; blue triangles in *Figure 2F*), comparable to that of the BCs and Bistrat cells. Apart from in vitro reports (*Hajos et al., 2013*), to our knowledge, this is the first in vivo evidence of a sub-population of AACs enhancing their discharge probabilities during ripple oscillations, with significant implications for GABAergic regulation of PC discharges during fast network oscillations (see 'Discussion').

Interestingly, the E-AACs showed preferred phases of firing during ripples comparable to Bistrat cells (*Figure 2F*; mean phase values for the E-AACs: 115° and 124°; note that the E-AACs also showed similar phase values to Bistrat cells during the epsilon oscillations as well; *Figure 2C*). Additionally, it is worthwhile to note that while only one of the AACs decreased its firing during the ripples, 5 out of the 5 C-AACs and 1 out of the 2 E-AACs showed significantly decreased probability of firing in the immediate post-ripple period (see *Figure 5G*). Similar post-ripple decrease in firing was reported previously for C-AACs from anesthetized rats (*Klausberger et al., 2003*).

Therefore, these observations demonstrate that there is a functional splitting of the AAC class during fast network events as well, in addition to the differences described above for the ripple-related discharges of the Bistrat cell sub-classes.

## Intra-class differences between BCs during fast network oscillations

Finally, we investigated whether intra-class differences in fast network event-related discharges could be identified for BCs as well. As a potential clue, we focused on the morphological differences identified above for the differentially discharging Bistrat cell and AAC sub-classes, namely, the presence of dendrites in the stratum radiatum vs oriens only (Bistrat cells) and the soma positions within or outside the PC layer (AACs). Indeed, BC somata have been reported to be occasionally located outside the stratum pyramidale (*Pawelzik et al., 2002*), but no functional differences have been reported between BCs situated inside vs outside the cell layer. Among our n = 12 successfully recorded and post-hoc visualized BCs, there were three cells with cell bodies and dendrites exclusively in the stratum oriens (*Figure 6B*). All of these three cells were positive for PV (*Figure 6E*) and targeted somata (*Figure 6F*; number of boutons examined that synapsed on somata: n = 9; dendrites: n = 3; target unidentified: n = 2). We refer to these BCs with somata and dendrites in the oriens as horizontal or H-BCs (note that H-BCs had axonal projections into [n = 2] or towards [n = 1] the subiculum). The H-BCs were clearly distinct from their more canonical counterparts (referred to as classical or C-BCs below), whose somata were located within the PC layer and the C-BCs had vertically oriented dendrites that spanned across the oriens to the radiatum, with branches extending into the lacunosum-moleculare layer (*Figure 6A*). These cells also showed PV immunoreactivity (*Figure 6C*) and targeted somata (*Figure 6D*).

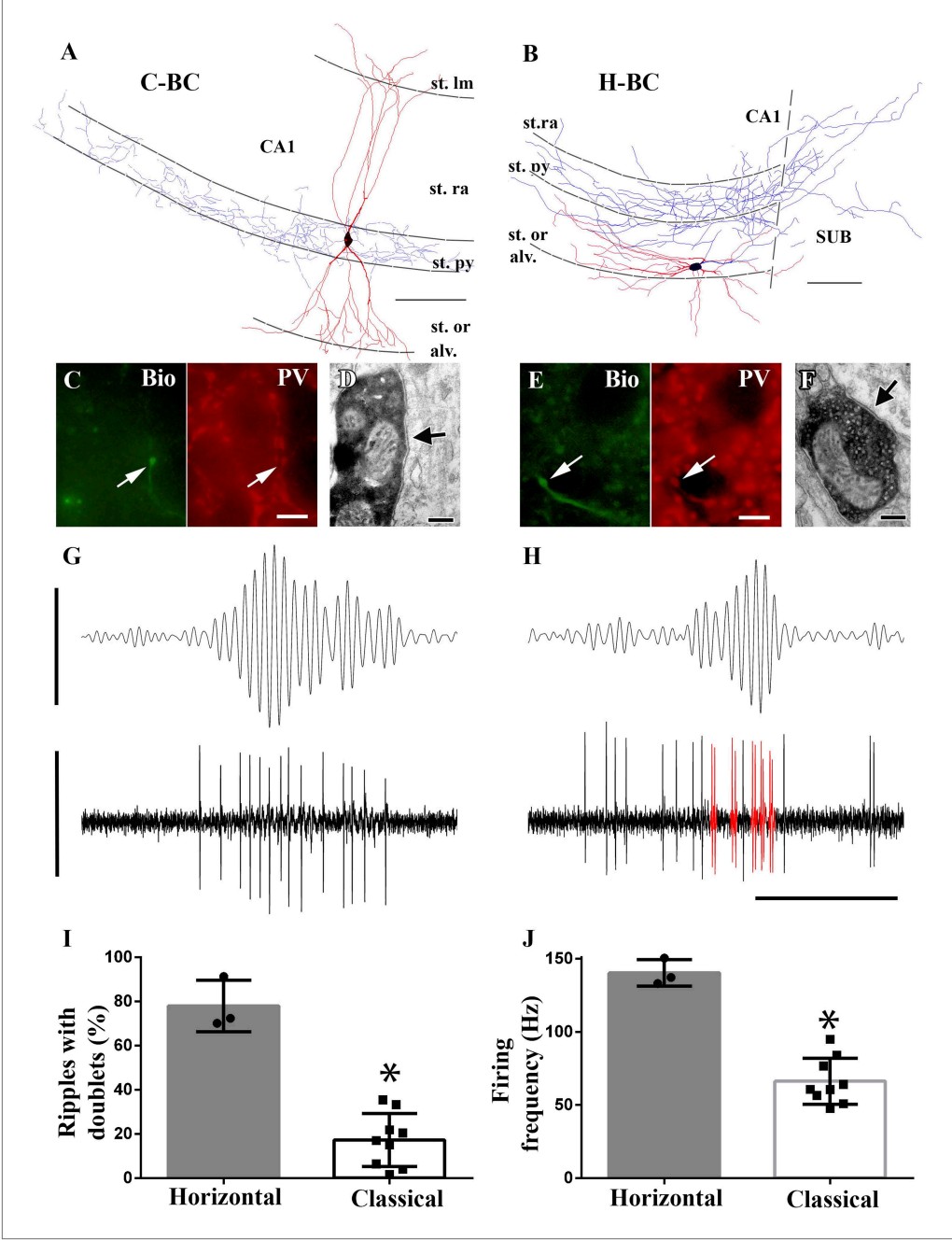

**Figure 6**. Horizontal-BCs located outside pyramidal layer fire doublets during sharp-wave associated events. (**A** and **B**) Full dendritic (red) and partial axonal reconstructions of a classical (**A**) and a horizontal (**B**) BC. Axons were partially reconstructed from single 60 μm sections for better visualization. Note that the H-BC had dendrites only in oriens/alveus, whereas the C-BC had dendrites in the stratum radiatum (and lacunosum-moleculare) as well. The H-BC sent axons both to the subiculum and CA1, in contrast to the C-BCs that innervated only the CA1 pyramidal layer. Scales: 100 μm. (**C** and **E**) Immunohistochemical verification of PV expression of cells in (**A**) and (**B**), respectively. Arrows point to neurobiotin (Bio) filled boutons (green) in the stratum pyramidale and to the corresponding PV positive profiles (scales: 10 μm). (**D** and **F**) Electron micrographs of boutons of the same classical (**A** and **D**) and horizontal (**B** and **F**) BCs innervating somata in the CA1 pyramidal layer. Arrows indicate the synaptic specializations. (**G** and **H**) Example band-pass filtered LFPs (90–200 Hz; upper traces) and the action potentials of the H- and C-BCs (**A** and **B**, respectively). Note that the H-BC repeatedly fired two spikes per oscillatory cycle (doublets: red), whereas the C-BC fired one spike per ripple cycle. LFP: 0.4 mV; unit traces: 1 mV; time scale: 100 ms. (**I**) The relative

*Figure 6. Continued on next page*

*Figure 6. Continued*

number of ripple events with doublets of action potential firing was significantly higher for H-BCs compared to C-BCs. (**J**) Average firing frequencies during ripple events were also significantly higher in H- than in C-BCs.

Importantly, analysis showed that the morphologically distinct BC sub-classes displayed differential firing behavior during fast network oscillations. Namely, H-BCs often fired two spikes per individual ripple cycle (i.e., doublets of action potentials) (*Figure 6H*), which occurred only rarely in C-BCs (*Figure 6G*). In addition, H-BCs also fired more intensely during ripples (*Figure 6H*). Quantification of the firing patterns of the two BC sub-classes showed statistical differences both in terms of the proportion of ripples when the cells fired doublets of action potentials (*Figure 6I*; H-BCs: 77.9 ± 11.6%; C-BCs: 17.3 ± 12%; p = 0.009, Mann–Whitney) and in terms of the average firing frequency during the ripples (*Figure 6J*; H-BCs: 128.9 ± 23.8 Hz; C-BCs: 62.7 ± 12.3 Hz; p = 0.004, Mann–Whitney).

Taken together, these results showed that intra-class differences in firing behavior during fast network oscillations also existed for the most numerically dominant PV interneuronal class, the BCs, similar to what was described above for the Bistrat cells and the AACs.

## Discussion

### Domain-specific temporal sequencing of GABA release during the descending phase of theta oscillations from PV interneurons

In this study, we examined the firing patterns of anatomically identified, fast-spiking, PV interneurons in the CA1 region of the mouse hippocampus under awake head-fixed conditions in order to provide insights into how the spatio-temporal patterns of GABA release from PV interneurons may regulate the excitability of populations of PCs during brain state-dependent hippocampal rhythms. Given that the canonical property of PV cells is their uniquely rapid signaling capability (*Norenberg et al., 2010*; *Armstrong and Soltesz, 2012*; *Chiovini et al., 2014*), our focus was on the fast (>90 Hz) oscillations. Since the epsilon rhythm occurs embedded within the theta oscillations, we first analyzed the discharge patterns of PV interneurons during theta waves. The data showed that the AACs fired with the strongest theta modulation. Unlike AACs from anesthetized rats (*Klausberger et al., 2003*), AACs in our awake head-fixed mice clearly did not fire with the highest probability close to the peak of theta rhythm (180°), but in fact discharged at 251° (the 71° difference corresponds to a 25 ms delay for an average, 8 Hz theta with a cycle duration of 125 ms). Although inter-species differences cannot be ruled out, the one AAC previously reported from the CA1 of non-anesthetized freely moving rats in the literature also discharged well after the theta peak (at 225°) (*Viney et al., 2013*). Furthermore, it is interesting to note that all of the 20 optogenetically tagged PV cells in a recent study conducted in freely moving mice discharged during the descending phase of the theta rhythm and none fired preferentially at the peak (*Royer et al., 2012*). Therefore, it appears that in animals during running, the axon initial segments of PCs receive a barrage of GABA inputs from AACs during the descending phase of the theta cycle (at 251° on average), followed by inputs from BCs onto their cell bodies and proximal dendrites about 20 ms later (at 310°), and then, with another approximately 20 ms delay on average, near the theta trough (360° or 0°), Bistrat cells release GABA onto the basal and mid-level apical dendrites. Thus, the pattern of discharges by the three cardinal PV interneuron classes is organized to generate a spatio-temporally exquisitely orchestrated axon initial segment to somata/proximal dendrites to mid-level dendrites sweep of fast GABAergic inhibition in a relatively short time window (*Figure 7A*, 'Theta').

### BCs, Bistrat cells, and AACs differentially discharge during fast network oscillations in the CA1 of awake mice

In terms of fast rhythms, our results revealed clear distinctions between the three major PV cell classes. First, BCs clearly differed from Bistrat cells and AACs in their strong modulation by epsilon oscillations. These results indicate that somatic and proximal dendritic GABAergic inputs from BCs may play primary roles in shaping the excitability of PCs during epsilon-associated behaviors. Second, BCs as a group also differed from the Bistrat cell and AAC classes in exhibiting the highest rates of firing during ripple events. Third, we found marked differences between Bistrat cell and BC discharges during ripples with low core frequencies, indicating that Bistrat cell-derived dendritic inhibition is only recruited

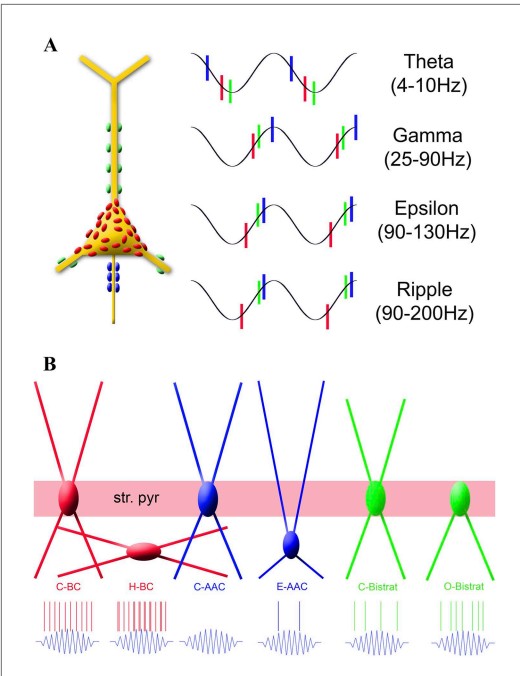

**Figure 7**. Summary of the temporal ordering of discharges by the three main PV cell classes (**A**) and the firing properties of the new sub-classes of PV cell during ripples (**B**). (**A**) Schematic illustration shows a PC with axon terminals from AACs (blue), BCs (red), and Bistrat cells (green); the same color code is used to illustrate the phase-preferential firing of the three major PV cell classes during theta, gamma, epsilon, and ripple oscillations. Note that the axon initial segment receives the earliest inputs during theta waves, whereas during the faster rhythms (gamma, epsilon, ripple) BCs fire first, followed by the Bistrat cells and the AACs. (**B**) Layer-specific locations of somata and dendrites segregate with the ripple-related discharge properties of the 6 PV sub-classes. Note the enhanced firing by the H-BCs, E-AACs, and the O-Bistrat cells compared to their classical counterparts (C-BCs, C-AACs, and C-Bistrats). DOI: 10.7554/eLife.04006.010

during high-frequency sharp-wave ripple events. Fourth, our data revealed that BCs on average fired earlier than Bistrat cells and AACs during all network oscillations faster than the theta range, including the running-associated, theta-embedded gamma and epsilon waves, as well as sharp-wave ripples during rest (**Figure 7A**). The latter finding indicates a behavioral state-independent leadership position (referred to as primacy below) of BCs in interneuronal firing sequences during fast oscillatory cycles, extending our previous observations of a frequency-invariant temporal ordering of BC and oriens-moleculare interneuron (OLM cells project to the distal dendritic regions of CA1 PCs) discharges, where the preferred phases of firing of BCs were also significantly earlier than the mean phase of discharges by the OLM cells during both epsilon and ripple events (**Varga et al., 2012**).

In addition to these four major inter-class differences during fast network oscillations, our data also showed that AACs in awake mice, when considered together as a single group, exhibited no significant changes in firing rates during ripple events, fired the majority of their spikes at the beginning of the ripples, and decreased their firing in the immediate post-ripple period, in overall agreement with data from anesthetized rats (**Klausberger et al., 2003**). Therefore, AACs in the CA1 do not necessarily stop delivering GABA during ripple events when sub-sets of PCs are known to replay previously experienced firing sequences (**Wilson and McNaughton, 1994**; **O'Neill et al., 2008**, **2010**), indicating that AACs may actively regulate PC discharges during sharp-wave ripples. The precise nature of the AAC to PC interaction in general (**Howard et al., 2005**; **Szabadics et al., 2006**; **Glickfeld et al., 2009**), and during ripples in particular, will have to be investigated further in future studies.

## Fast network oscillations reveal significant intra-class differences between members of all three PV classes that segregate with morphological properties

Importantly, in addition to inter-class differences between PV cells, our data also revealed that robust intra-class distinctions emerged from all three major PV cell classes during fast (>90 Hz), but not slow, network oscillations (**Figure 7B**). First, Bistrat cells without dendrites in the stratum radiatum, the O-Bistrat cells, fired at significantly higher rates during ripple events than their C-Bistrat counterparts. These closely correlated morphological and physiological differences firmly establish the O-Bistrat and C-Bistrat cell groups as distinct interneuronal sub-classes and highlight the fact that intense action potential discharges during sharp-wave ripples can be generated by interneurons without dendrites in the stratum radiatum.

Second, E-AACs were significantly modulated by epsilon and ripple waves, in contrast to their counterparts located in the stratum pyramidale. In addition, E-AACs showed significantly elevated frequency of action potential discharges during ripples. Third, H-BCs fired more intensely during ripples and discharged more doublets of action potentials during individual ripple cycles than the C-BCs. Differences in somatic layer position, in conjunction with significant physiological differences, qualify

these distinct AACs and BCs as separate, novel interneuronal sub-classes (note that layer-specific location together with differential spiking behaviors in vivo was also used as the basis of the differentiation of a previously unified cell class into neurogliaform and ivy cell sub-classes [*Fuentealba et al., 2008*]). Importantly, these newly identified differentially discharging PV sub-classes are likely to be not negligible in terms of their abundance, as the population of PV interneurons in the stratum oriens has been estimated to be about 30% of the number of PV cells within the stratum pyramidale (in 1000s: BCs: 1.3/3.9; Bistrat cells: 0.5/1.6; AACs: 0.3/1) (*Bezaire and Soltesz, 2013*). Reassuringly, the H-BC/C-BC and E-AAC/C-AAC ratios (3/10 and 2/5, respectively) in our sample approximated 30%, indicating that we did not under-sample the oriens layer cells.

## Primacy of BCs as a potential factor contributing to the differential discharges of PV cell sub-classes during fast events?

Future studies will be needed to identify the reasons for the differences in firing frequencies of these new PV sub-classes during fast rhythms in the CA1, but they may include potential differences between intrinsic excitability (e.g., as reported for fast spiking BCs and AACs in the neocortex and CA3) (*Woodruff et al., 2009*; *Papp et al., 2013*) and excitatory and inhibitory connectivity (*Hajos et al., 2013*) both from within the CA1 as well as from extrinsic sources (e.g., AACs have been reported to receive differential inhibitory innervation from the septum [*Viney et al., 2013*]). Among the circuitry related differences that may contribute to the functional splitting of the PV classes during fast oscillations, it is interesting to consider the potential role of the primacy of the BCs discussed above. What consequences would the consistently early firing of BCs during cycles of fast network oscillations have on the excitability of the other PV cells? One clue to this question may be related to the fact that BCs are known to synapse not only on PCs but also on the other PV expressing cells as well (*Cobb et al., 1997*). A second potential clue is that BC axons are largely confined to the stratum pyramidale (*Glickfeld and Scanziani, 2006*; *Foldy et al., 2007*), thus, PV cells whose cell bodies and proximal dendrites are located outside the PC layer will be in a privileged position to escape inhibition from BCs. A third relevant point is that our data showed that the average phase-preferential discharges of Bistrat cells, and the two epsilon and ripple-modulated AACs in our sample, were delayed by about 1.4 ms (calculated for a 140-Hz ripple) with respect to the mean phase of firing by BCs during fast network events, a period of time that is approximately a monosynaptic delay. Therefore, firing of BCs early during each ripple cycle is expected to deliver synaptic inhibition to other BCs as well as to AACs and Bistrat cells. The inhibition to other BCs would arrive after the BCs fired their first action potentials, thus, it can only prevent the firing of subsequent action potentials by BCs during each individual ripple cycle. Indeed, although BCs are known to be able to fire at 100s of Hz with little or no accommodation (*Kawaguchi, 1995*; *Buhl et al., 1996*), most BCs fired only a single spike per ripple cycle. The exceptions were precisely those BCs whose cell bodies were outside of the PC layer (and thus largely outside of the BC axonal cloud) the H-BCs, which discharged at higher intra-ripple frequencies and fired frequent doublets during ripple cycles (*Figure 7B*). The early firing of BCs is expected to inhibit the later-discharging AACs and Bistrat cells as well, likely contributing to their relatively lower frequencies of firing during ripples. However, AACs whose cell bodies are outside of the PC body layer would be predicted to be able to largely escape the BC-derived inhibition, which is consistent with our observation that such E-AACs significantly increased their firing rates during ripple events (*Figure 7B*). While future studies will be needed to identify the precise factors shaping the temporal ordering of interneuronal discharges, the early firing of BCs during fast network events is likely to differentially regulate the excitability of those interneurons whose cell bodies are located inside vs outside the termination zone of BC axons.

## Limitations of the study

A particular strength of our study is the rigorous identification of the recorded fast-spiking interneurons. However, the juxtacellular in vivo technique is based on non-targeted ('blind') recordings from actively discharging interneurons, and even after successful post-hoc visualization and cell identification it produces only relatively limited sample sizes. Nevertheless, even though only a single cell was attempted to be juxtacellularly labeled in each animal, the database of PV cells presented in this paper is the largest collection of anatomically and immunocytochemically identified fast-spiking cells that have been recorded from non-anesthetized animals in vivo to date. Interneurons are a minority (estimated to be 38,500/349,500 or about 11%) of the total CA1 neuron population, and PV cells comprise approximately a quarter of the interneurons in the CA1 (9200/38,500, 23.9%) (*Bezaire and Soltesz, 2013*). Among the

PV cell classes, BCs are estimated to be the most numerous (14% of all CA1 interneurons), followed by the Bistrat cells (6%) and the AACs (4%) (*Baude et al., 2007*; *Bezaire and Soltesz, 2013*). Reassuringly, these frequencies of occurrence roughly matched the ratio of the cell numbers for the three PV cell classes in our database (n = 12 BCs; n = 8 Bistrat cells; n = 5 AACs), indicating a lack of a major sampling bias for the various PV cell classes in our juxtacellular recordings.

Although recordings from head-fixed animals have certain limitations (e.g., in studies of eye movements) (*Wallace et al., 2013*), an advantage of the head-fixed juxtacellular recording approach (*Varga et al., 2012*) for the study of network oscillations is that the mice under head-fixed conditions produce relatively long periods of running (typically tens of seconds), which allows robust sampling of continuous theta rhythm and the associated theta-nested epsilon oscillations while maintaining high-quality juxtacellular recordings from single cells. In contrast, theta-related juxtacellular recordings from non-head-fixed freely moving but tethered and spatially confined rats were reportedly limited to short periods of head movements and postural shifts while the animals remained in the same location (*Viney et al., 2013*), highlighting the existence of both advantages and disadvantages of the particular experimental arrangements.

An important conclusion from our study is that caution has to be exercised when trying to infer precise interneuronal identity from extracellular unit data, even if combined with optogenetic tagging. Nevertheless, our results also highlight new criteria that may now be employed. For example, although a hypothetical extracellularly recorded unit that increases its firing during sharp-wave ripples may involve any of the three major PV classes, BCs and AACs can be sharply distinguished based on the strength of epsilon modulation of their firing (note that all BCs were strongly epsilon modulated, whereas AACs were weakly or not significantly modulated). Similarly, Bistrat cells can be distinguished from BCs based on their phase-preferential firing during theta and ripples, in combination with their modulation during epsilon waves. Such insights are important in light of the fact that numerous studies have been conducted using extracellular unit recordings from freely moving animals using tetrodes and silicon probes, and post-hoc analysis of existing data may now be attempted to dissect the role of specific PV cell populations in particular behavioral tasks.

## Functional relevance and outlook

The existence of three new PV cell sub-classes in the CA1 region beyond the classically identified monolithic BC, Bistrat cell, and AAC groups is likely to have major consequences for the mechanisms by which local GABAergic fast inhibition regulates output from the hippocampus. In particular, our results demonstrating a functional splitting of the three major PV cell classes may serve to increase the available repertoire of inhibitory regulatory processes to facilitate the coordination of PC excitability during fast network rhythms associated with various cognitive and behavioral processes (*Staba et al., 2004*; *Mukamel et al., 2005*; *Canolty et al., 2006*; *Tort et al., 2008*; *Colgin et al., 2009*; *Kepecs and Fishell, 2014*). The preferential ability to generate doublets of spikes by the distinct BC sub-populations identified in this study is particularly interesting from a functional perspective. Indeed, firing of doublets of action potentials by BCs has been suggested to play key roles in synchronization of spatially distinct neuronal sub-populations during gamma rhythm (*Traub et al., 1996*). Therefore, it will be important to determine if the doublet firing that we found preferentially in the H-BCs serves a similar function during ripples.

The hippocampus plays key roles in learning and memory processes by transforming input from associative neocortical regions to outputs carried by the long-distance projecting axons of CA1 PCs targeting a variety of brain areas, including the medial entorhinal cortex, amygdala, and the medial prefrontal cortex (*Cenquizca and Swanson, 2007*; *Lee et al., 2014*). Interestingly, recent results demonstrated that CA1 BCs differentially innervated sub-populations of CA1 PCs, both in terms of their superficial vs deep positions within the stratum pyramidale and with regards to their long-distance projection targets (*Lee et al., 2014*), likely contributing to the sparse and distributed nature of hippocampal network activity. Since a high degree of bias was also noted in the excitatory innervation of BCs by the distinctly projecting CA1 PCs as well (e.g., medial prefrontal cortex-projecting PCs were much more likely to innervate BCs than amygdala-projecting ones) (*Lee et al., 2014*), it will be especially important to determine if such specialized local inhibitory–excitatory circuits also involve AACs and Bistrat cells in general and the newly recognized PV sub-classes in particular.

Our study also highlights the importance of future investigations into the developmental origins of the distinct BC, Bistrat cell, and AAC sub-classes. Indeed, functionally distinct, new sub-classes of OLM cells, recently identified based on expression of 5HT(3A) receptor, differentially participated in

network oscillations and originated from distinct sub-divisions of the ganglionic eminences during embryonic development (*Chittajallu et al., 2013*). Therefore, developmental mechanisms may drive the formation of specialized microcircuits that can parse the various CA1 PC sub-populations according to the dynamically changing circuit demands during hippocampal memory functions.

## Materials and methods

All experiments were approved by the Institutional Animal Care and Use Committees of the University of California at Irvine, the University of California at Los Angeles and the University of Pecs. Anesthesia, surgical procedures, experimental design, and data acquisition were the same as described earlier (*Varga et al., 2012*). Briefly, head bars were implanted during deep isoflurane anesthesia on C57BL/6 male and female adult (3–12 months) mice. The animals were acclimated to run on an 8-inch spherical treadmill. On the day of data collection, craniotomy was performed over the dorsal hippocampus (−2 mm AP and 2 mm lateral) under deep anesthesia (isoflurane). Recordings were started after the animal became fully alert (min 30 min after termination of anesthesia), and the two glass recording electrodes (borosilicate, long taper) were placed into the stratum pyramidale (LFP channel, for field potential recording; its position in the stratum pyramidale was carefully adjusted based on the maximal ripple amplitude) and into the stratum oriens/pyramidale for single unit recordings/juxtacellular labeling (referred to as the unit channel below). After neurobiotin labeling of the recorded interneurons (1.5–2% in 0.5 M NaCl in the juxtacellular electrode [*Pinault, 1996*]), the experiments were terminated and the animals were perfused within 4 hr under deep anesthesia. Fixatives contained 4% paraformaldehyde, 0.1–0.05% glutaraldehyde, 15% saturated picric acid, and 0.1 M phosphate buffer (PB, pH = 7.4). The brains were cut into 60-μm thick sections with a vibratome. Neurobiotin was first visualized with Alexa Fluor 488 conjugated streptavidin (Invitrogen, Carlsbad, CA) and samples including soma/dendrite/axons were selected for further immunohistochemical analysis. Specific protein expressions were checked with the help of the following antibodies: rabbit anti-PV (Swant, Switzerland), mouse anti-PV (Sigma-Aldrich, Carlsbad, CA), rat anti-somatostatin (Millipore, Temecula, CA), goat anti-SATB1 (Santa Cruz Biotechnology, Dallas, TX), goat anti-mGluR1a (Frontier Science Co., Ltd, Japan). Antibodies were diluted in 0.1 M PB containing 0.3% Triton-X100 and sections were incubated overnight at room temperature in the antibody cocktails. Secondary antibodies (raised in donkey against rabbit/goat/mouse/rat and conjugated to Dylight 594/649/405, Jackson Immunoresearch, USA) were used to detect the location of the primary antibodies. Documentation of the immunoreactivity was performed with either a CCD camera attached to a Zeiss Axioscope microscope or confocal microscopy with an Olympus FV1000 or Zeiss LSM780 microscope. Finally the neurobiotin filled cells were visualized with 3,3′-diaminobenzidine tetrahydrochloride (DAB), dehydrated and embedded for electron microscopy. Fully recovered cells were 3D reconstructed with Neurolucida (MBF Bioscience, Williston, VT) system using a 100× objective (N = 1.3). Lengths of dendrites were measured only on completely reconstructed dendritic arborizations.

Cells were identified based on their protein expression and/or axonal targets. Somatostatin immuno-reactivity was confirmed on all eight Bistrat cells. The axon of one Bistrat cell was not filled with neurobio-tin sufficiently, thus we tested the immunopositivity of this cell to mGluR1alpha, which is a negative marker for Bistrat cells, but present in other somatostatin-positive interneurons (*Baude et al., 1993*). Synaptic targets of AACs (4 out of 5) and BCs (4 out of 12) were verified on random samples collected from the stratum pyramidale and proximal oriens/radiatum on ultrathin (70 nm) sections. Images were taken with a Gatan SC1000 ORIUS CCD camera attached to a Jeol (JEM-1400) electron microscope.

## Analysis

Minimal duration of episodes for analysis was >20 s for running and >200 s for resting episodes. In the case of two cells (imi069 and imi075 in *Table 1*), mice had been injected with saline solution (50 nl) in the amygdala at least 2 weeks before recording as part of a control group in an on-going parallel study; the firing properties of these cells did not differ from their counterparts recorded from non-injected mice, and thus they were included in the database. The analysis was performed with the help of custom-written Matlab scripts as described earlier (*Varga et al., 2012*) with some additional features/modifications described below. The low-pass filtered (Bessel, 5 kHz) signals of the two adjacent channels were sampled at 12 or 20 kHz and stored for offline analysis. Running-associated theta was filtered (5–10 Hz) on downsampled (20 times) signal of the unit channel, high-frequency oscillations were analyzed on the LFP channel. Running-associated LFP signals were filtered for epsilon oscillations between 90–130 Hz. For high-frequency oscillatory epochs during resting (ripples), LFP was filtered between

90–200 Hz. First, envelope-detection was performed on the absolute values of the signals, and instances where the envelope crossed the threshold of 5 standard deviations were considered as ripples. The start and end of ripples were specified as the time points where the signal reached a second threshold of 2 standard deviations. For display purposes (*Figures 2E and 5G,H*), before and after ripple periods (lasting two times the duration of the ripples) were also included, and spiking probabilities were binned into 40, 20, 40 bins corresponding to before, during, and after ripple periods, respectively.

Time–frequency plots were generated on the LFP channel signal. First, the signals were narrow-band filtered (4 Hz bandwidths from 10–200 Hz, 2 Hz steps [*Canolty et al., 2006*]), then envelope-detection was performed on the absolute values of the signals and the power of the envelope was normalized in every frequency band. Theta trough or ripple peak power triggered averaging of the resulting data was performed and displayed for cross-frequency coupling during running-associated theta and for resting-associated ripple activity, respectively.

Spike occurrences during the various oscillations were determined and uniformity was tested with Rayleigh-test (*Varga et al., 2012*). Preferred firing phase and strength of modulation during oscillations of individual cells were calculated as described earlier (*Varga et al., 2012*) and expressed as orientation and length of vectors calculated by summing the spike phases normalized by the number of spikes. Depending on the orientations of the different phase angles, the length of the normalized vector (r) ranged from 0 (no phase preference) to 1 (all phase angles identical). This length of the vector was used to measure the magnitude of phase modulation, and the direction of r indicated the mean phase angle of the cell (*Varga et al., 2012*). Differences between cell classes (BCs, Bistrat cells, AACs) and sub-classes (*Figure 7B*) in terms of phase locking were evaluated using Watson–Williams circular test (*Senior et al., 2008*). In order to determine whether the firing probability of a given cell was significantly lower or higher during ripples than outside ripples, simulations were performed either by randomizing the spike location (*Varga et al., 2012*) or by shuffling the ripple location (*Lasztoczi et al., 2011*).

The core frequency of ripple epochs was calculated with three different methods. The first method used the inverse of the averaged times between zero crossings of the ripple signal within the ripple boundaries and divided by two. The second method computed the inverse of the time between neighboring peaks of ripples within the ripple boundaries (*Tukker et al., 2013*). The third method generated wavelet transforms on the 90–200 Hz filtered LFP signal using 'cmor5-1' function in the Matlab Wavelet Toolbox. The peak values of the wavelet transforms were measured and counted as the core frequency (*Patel et al., 2013*). All three methods were tested on artificial ripple events and found to show similar core frequency detection. We report our data throughout the manuscript with the wavelet method. For the analysis associated with *Figure 3*, events detected on the 90–200 Hz band-pass filtered LFP signals during rest were sorted based on the presence or lack of action potentials of the recorded Bistrat and BC cells. Core frequencies of events with or without spikes were statistically compared with Wilcoxon signed rank test. With the exception of the inset in *Figure 3D*, data are presented as mean ± standard deviation.

## Acknowledgements

We thank R Zhu and N Mike for technical assistance, DC Lyon for help with Neurolucida System.

## Additional information

### Funding

| Funder | Grant reference number | Author |
| --- | --- | --- |
| National Institutes of Health | NS35915 | Ivan Soltesz |
| U.S. Department of Veterans Affairs | 1I01BX001524-01A1 | Peyman Golshani |
| Epilepsy Foundation | | Csaba Varga |
| Hungarian Academy of Sciences | National Brain Research Program KTIA_NAP_ 13-2-2014-0003 | Csaba Varga |
| George E. Hewitt Foundation for Medical Research | | Gergely G Szabo |

The funders had no role in study design, data collection and interpretation, or the decision to submit the work for publication.

## Author contributions

CV, Designed and performed experiments and analysis, Wrote the manuscript; MO, MB, Performed analysis; JL, Performed experiments and analysis; GGS, IM, Performed experiments; PG, IS, Designed experiments, Wrote the manuscript

## Ethics

Animal experimentation: This study was performed in strict accordance with the recommendations in the Guide for the Care and Use of Laboratory Animals of the National Institutes of Health. All procedures were approved by the University of California Irvine Animal Care and Use Committee (protocol #1999-1719) and the University of Pecs, Hungary. All surgery was performed under deep isoflurane anesthesia, and every effort was made to minimize suffering.

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
