## [Decision Letter]

Thank you for sending your work entitled “Functional Fission of Parvalbumin Interneuron Classes During Fast Network Events” for consideration at *eLife.* Your article has been favorably evaluated by Eve Marder (Senior editor) and 3 reviewers, one of whom is a member of our Board of Reviewing Editors.

The Reviewing editor and the other reviewers discussed their comments before we reached this decision, and the Reviewing editor has assembled the following comments to help you prepare a revised submission.

Overall, the reviewers felt that the work represented a novel and significant contribution but several issues were raised that need to be addressed. In general, additional analyses and editing are required. Please note that while additional experiments are not specifically being requested in the revision, helpful further experiments have been indicated in specific comments below, and major revisions are being suggested.

First, while there was concern about the small numbers of cells, it was considered reasonable given the technical challenges associated with the work, and given that small numbers of cells were also presented in previously published work with anesthesized animals. It might be helpful to point this out in your revised submission, and also to provide more complete information on the given cells, perhaps in a table (see 1st point of reviewer#1, and 1st point of reviewer#2). Second, it was felt that additional analyses needed to be done which considered gamma (see 3rd point of reviewer#3 below) and in general, clarity and explanation about frequency ranges of analyses (see 3rd point of reviewer #2, 2nd point of reviewer #3), as well as wording consistency (see 1st point of reviewer#2). It is suggested that the authors could start by specifying the ranges considered based on what criteria (what previous works, why etc.). It is recognized that it is confusing in the literature and people use different ranges and terminology for theta, gamma, ripple etc., but the reviewers urge the authors not to add to the confusion in the literature, and present analyses accordingly. Third, the reviewers request that more explanations be provided in the revised work, and that it would be helpful to include schematic figures to summarize results and speculations presented (see 2nd and 3rd points of reviewer#1, 1st and 2nd points of reviewer#2 and first point of reviewer#3).

*Reviewer #1*:

The challenge of interneuronal heterogeneity is immense, and a recent review by [44] suggests that we need to consider interneuron diversity from functional perspectives (behaviour or network dynamics), rather than simply classification. In this work, the authors do this, considering phase and firing of identified interneurons during oscillatory activities in vivo. Juxtacellular recordings are used in head-fixed mice, allowing long, stable recordings. From this perspective, sub-classes of fast-spiking PV+ interneurons are noted with particular morphological differences. The results are clearly described and the long discussion is helpful in putting the work in perspective. However, I have some comments.

1) Given the limited number of recordings, it would be helpful to provide more detail in some places. For example, it was noted that 3 out of 5 bistrat were PV+ (immuno). What about the other 2? Technical issue? Which of these are the O-Bistr and C-Bistr? Perhaps Tables detailing all the data technical aspects could be done.

2) I could not find the meaning of 'modulation strength'? How was this calculated? Again, because the numbers are limited, it would be helpful to be clear and specific about all calculations done, and modulation strength is critical in this work, so wondering if it is possible whether the way the calculation was done may be biased in any way given in the limited data set.

3) Although I recognize the challenge, and the combination of interneuron identification and firing properties during different frequencies/behaviours (run/rest) is admirable, the paper at times felt a bit like 'exploratory observations'(albeit interesting observations), since there is not an overall mechanism of how the oscillations are produced and why and how the firing and phasing might be as such. The authors bring up interesting perspectives that their data support, e.g., “...the pattern of discharges by the three cardinal members of the PV interneuron family is organized in such a way that it generates a spatio-temporally exquisitely orchestrated, initial segment to somata/proximal dendrites to mid-level dendrites sweep of fast GABAergic inhibition in a relatively short time window.”

However, if possible, it would be useful to perhaps put forth a speculative mechanistic picture of the overall population oscillation and cellular activities.

*Reviewer #2*:

The study by Varga and colleagues shows an unexpected diversity of PV expressing interneurons in the CA1 hippocampal subfield in terms of the location of their cell bodies, dendritic arbours and their discharge activity during fast (> 90 Hz) network oscillations. One important and new observation is, that one subpopulation of PV-expressing basket cells (BCs) with soma location in the stratum oriens generates action potentials during ripple activity at higher frequencies than the remaining PV-BC types. The authors argue that this sub-population is in the privileged position to escape from periosmatic inhibition provided by other PV-BCs with soma and axon in the principal cell layer and thereby provide perisomatic inhibition to the CA1 circuitry, including other PV-interneurons. The authors further argue that this is reflected in the delay of only 1.4 ms between the discharge of the privileged PV-BC sub-population and the other PV cells. Although this prediction is indeed acceptable, its functional significance or relevance on network function or encoding of information is not well addressed and leaves the reader with the open question 'what is it good for?' The other new finding of the study is that a sub-population of axo-axonic cells (AACs) with soma location in the stratum oriens discharge during ripple activity.

This finding stays in contrast to previous investigations from Klausberger et al. (2003; Science) showing a lack of activity of AACs during ripple oscillations. This is an interesting observation; however, a major concern is the low number (two) of recorded and identified AACs which form the basis of this conclusion. A generalization of morphologically and functionally driven sub-classifications of AACs is therefore highly speculative. A further main and more general criticism relates to the quality of the written results section. Too many text passages are included in brackets which could easily be included in the flow of the main body of the text. Makes reading a bit clumsy. Moreover, inconsistencies in the usage of abbreviations and not necessary figure legends included in the main body of the text makes the reading of the manuscript at its present stage less accessible.

1) The terminology 'family' of interneurons, 'classes' and sub-classes' is not defined making it difficult to understand from which group of cells the author is referring to. A clear definition at some point is therefore needed. More critical is however the sub-classification of AACs with soma location in stratum oriens on a very small sample (2 cells; see above). In order to improve the validity of the subdivision of AACs, a higher number of identified cells would be urgently needed. The observed difference in their activity in comparison to AACs with axon in the principal cell layer during ripple oscillations indicates that they receive different excitatory inputs. This is discussed by the authors. A parallel set of experiments (in vitro) would be helpful to directly test this hypothesis and to prove whether the two subsets of AACs receive different total excitatory inputs (spontaneous EPSCs, evoked responses). Finally, additional values on the intrinsic properties of the two AAC subclasses as well as more detailed morphological characteristics such as total dendritic and axonal length as provided for the two bistratified cell sub-classes would be important to possibly strengthen the proposed functional sub-classification. This could be done under in vitro conditions.

2) Although BCs with somata in the stratum oriens fire earlier than the other PV-cells and therefore might, as suggested by the authors provide perisomatic inhibition onto the other PV cells which finally may result in the observed 1.4 ms delay, the functional relevance of this delay in respect to encoding of information remains open. One explanation regarding the sub-classes in respect to PC innervation is that they may innervate different sub-classes of PCs (superficial versus deep PCs). This is an interesting hypothesis which the authors could have addressed in order to strengthen the functional significance of the various interneuron sub-classes. Again, this could be done under in vitro conditions.

3) It is unclear and confusing that the upper gamma frequency range (90-130 Hz) is determined epsilon and not 'upper gamma'. The terminology 'upper gamma' is broadly used in the literature and corresponds in its relation to the theta phase exactly to the 'epsilon' activity of this study.

*Reviewer #3*:

In this work, the authors perform juxtacellular recordings of parvalbumin positive interneurons near the CA1 pyramidal layer of the mouse hippocampus in head-fixed unanesthetized mice running and resting on a sphere during pyramidal layer theta, epsilon and ripple oscillations. Using these data, they make a number of novel observations regarding the firing properties of these cells. The study itself is important, for the large juxtacellular recording sample of PV interneurons in an unanesthetized preparation, and for providing valuable data regarding the importance of morphological details in determining the firing properties of interneurons. However, there were also some issues.

One of the claims in the current manuscript is that AAC cells were not inhibited during ripples. In fact, none of the interneurons recorded in this or an earlier study (91) shows any ripple-inhibited neurons. Yet, ripple-inhibited interneurons are regularly observed in freely-moving rats, so this difference cannot be explained by anesthesia, which is one of the possibilities provided. For example, Csicsvari et. al. 1999 (which should be discussed in this context) shows anti-SPW cells in 13 out of 81 cells in the alveus/oriens. The present and earlier studies features mainly cells from the pyramidal layer, but also some from oriens. Could it be due to minimal sampling of oriens cells in this account? A more detailed and reasoned explanation is necessary to help understand why there is such significant deviation from previously published reports.

Importantly, the use of 90-200 Hz for the “ripple” oscillation is not standard, as the authors acknowledge, and likely problematic. Sullivan et. al., which is cited to justify this frequency range, noted that 90-140 Hz fast gamma oscillations are distinct from sharp-wave ripples. For one, fast gamma oscillations seem to be local and low-amplitude, which some sharp-wave ripples resonate along the entire septo-temporal axis. The pooling of these events could account for the previously mentioned discrepancy with other studies-perhaps some cells (including Bistrat cells) do not fire during fast gamma, but only ripples. Judging from the examples in Figure 3, the fast gamma (slow ripple) events appear to be of lower amplitude/power as well. This should be corrected to allow for proper evaluation and comparison of these results with other reports.

The authors did not perform any gamma oscillation analyses. The current account is incomplete as gamma is highly relevant for PV interneurons (see their 2012 PNAS study). For example, are the gamma phase preferences similar to epsilon phase preferences, or do they reflect similar temporal delays as in the earlier [91] study? It would also be interesting to know the gamma phase preferences for the AAC cells which do not fire during epsilon.

[Editors' note: further revisions were requested prior to acceptance, as described below.]

Thank you for resubmitting your work entitled “Functional Fission of Parvalbumin Interneuron Classes During Fast Network Events” for further consideration at *eLife.*

Your revised article has been favorably evaluated by Eve Marder (Senior editor), a Reviewing editor, and one of the previous reviewers. The manuscript has been substantially improved. However, we would appreciate the authors addressing the following two issues:

1) To support the lumping of the fast frequencies, please show a plot of the distribution of core frequencies for fast frequency events (> 90 Hz) to show that it is not a bimodal distribution composed of distinct fast gamma and ripple events (e.g. as reported by Sullivan et. al (Figure 1).

2) Since Panel G of Figure 5 has been significantly changed since the previous submission, please confirm that the xtic alignment is appropriate as it seems quite different from the previous version even considering the additional cells. It was noted that it appears that the ripple firing is in fact lower for these cells, particularly after the ripple onset. In Klausberger et. al.'s 2003 Nature paper the AAC cells were reported to fire biphasically during ripples. According to Klausberger et al, averaged across the ripple AAC cells “showed no significant difference in discharge frequency”, but they increased their firing at the onset of ripples, but were silent only after the ripple peak (see their Figure 4). So it may be that the data does not differ from that reported by Klausberger et. al., as stated in the Discussion section. Please address this.

---

## [Author Response]

*First, while there was concern about the small numbers of cells, it was considered reasonable given the technical challenges associated with the work, and given that small numbers of cells were also presented in previously published work with anesthesized animals. It might be helpful to point this out in your revised submission, and also to provide more complete information on the given cells, perhaps in a table (see 1st point of reviewer#1, and 1st point of reviewer#2)*.

We have included a new table (Table 1) that lists all recorded cells, specifying the cell types and the immunocytochemical and EM analysis used for the cell identification. In addition, we expanded and clarified the description of the cell identification strategies in Results.

*Second, it was felt that additional analyses needed to be done which considered gamma (see 3rd point of reviewer#3 below) and in general, clarity and explanation about frequency ranges of analyses (see 3rd point of reviewer #2, 2nd point of reviewer #3), as well as wording consistency (see 1st point of reviewer#2). It is suggested that the authors could start by specifying the ranges considered based on what criteria (what previous works, why etc.). It is recognized that it is confusing in the literature and people use different ranges and terminology for theta, gamma, ripple etc., but the reviewers urge the authors not to add to the confusion in the literature, and present analyses accordingly*.

As requested, we carried out an in-depth analysis of the PV interneuron discharges during gamma oscillations, and the results are included in the manuscript. In addition, we have included a clear rationale and explanation for the nomenclature and frequency ranges used for the analysis of the various oscillations. Regarding wording consistency, we eliminated all superfluous and ill-defined terms such as “family”, “groups/subgroups”, “types/subtypes”, and used only the terms “class” and “subclass” throughout the text.

*Third, the reviewers request that more explanations be provided in the revised work, and that it would be helpful to include schematic figures to summarize results and speculations presented (see 2nd and 3rd points of reviewer#1, 1st and 2nd points of reviewer#2 and first point of reviewer#3)*.

We have included a new summary figure (Figure 7) that highlights the key findings about the temporal sequences of phase-preferential firing by the three major PV cell classes and the ripple-related discharge patterns of the newly described PV interneuron subclasses.

Reviewer #1:

*1) Given the limited number of recordings, it would be helpful to provide more detail in some places. For example, it was noted that 3 out of 5 bistrat were PV+ (immuno). What about the other 2? Technical issue? Which of these are the O-Bistr and C-Bistr? Perhaps Tables detailing all the data technical aspects could be done*.

The new Table 1 lists the immunocytochemical and EM identification data for each cell. Regarding the Bistrat cells immunoreactivity, we included a sentence that addresses this issue.

*2) I could not find the meaning of 'modulation strength'? How was this calculated? Again, because the numbers are limited, it would be helpful to be clear and specific about all calculations done, and modulation strength is critical in this work, so wondering if it is possible whether the way the calculation was done may be biased in any way given in the limited data set*.

The method for calculations of modulation strength is now explained and it is also specified there that the method was identical to what was described in detail in [91].

*3) Although I recognize the challenge, and the combination of interneuron identification and firing properties during different frequencies/behaviours (run/rest) is admirable, the paper at times felt a bit like 'exploratory observations'(albeit interesting observations), since there is not an overall mechanism of how the oscillations are produced and why and how the firing and phasing might be as such. The authors bring up interesting perspectives that their data support, e.g., “...the pattern of discharges by the three cardinal members of the PV interneuron family is organized in such a way that it generates a spatio-temporally exquisitely orchestrated, initial segment to somata/proximal dendrites to mid-level dendrites sweep of fast GABAergic inhibition in a relatively short time window*.*”*

*However, if possible, it would be useful to perhaps put forth a speculative mechanistic picture of the overall population oscillation and cellular activities*.

We included a summary figure (Figure 7) that consists of two panels and related key messages. Panel A shows the preferential firing of the cell classes during theta, gamma, epsilon and ripple oscillations, visualizing the temporal sequencing of the GABA release at the axon initial segments, somata and dendrites. Panel B illustrates the firing probabilities of the 6 PV subclasses during ripples, including the doublets discharged by the H-BCs, highlighting the elevated ripple-related firing by the interneuronal subclasses located outside of the stratum pyramidale and the BC axonal cloud (H-BCs and E-AACs), and by the Bistrat cells that lack dendrites in the stratum radiatum (O-Bistrats). While at the moment we cannot offer a full mechanistic understanding of the patterning of discharges by the PV classes and subclasses during the various oscillations, we put forward a mechanistic hypothesis that proposes that cells with somata outside the stratum pyramidale escape inhibition from BCs and thus fire more than their counterparts in the stratum pyramidale. In addition, we think that our findings will inspire and guide future investigations into the mechanistic questions raised by our study.

Reviewer #2:

*1) The terminology 'family' of interneurons, 'classes' and sub-classes' is not defined making it difficult to understand from which group of cells the author is referring to. A clear definition at some point is therefore needed. More critical is however the sub-classification of AACs with soma location in stratum oriens on a very small sample (2 cells; see above). In order to improve the validity of the subdivision of AACs, a higher number of identified cells would be urgently needed. The observed difference in their activity in comparison to AACs with axon in the principal cell layer during ripple oscillations indicates that they receive different excitatory inputs. This is discussed by the authors. A parallel set of experiments (in vitro) would be helpful to directly test this hypothesis and to prove whether the two subsets of AACs receive different total excitatory inputs (spontaneous EPSCs, evoked responses). Finally, additional values on the intrinsic properties of the two AAC subclasses as well as more detailed morphological characteristics such as total dendritic and axonal length as provided for the two bistratified cell sub-classes would be important to possibly strengthen the proposed functional sub-classification. This could be done under in vitro conditions*.

We have eliminated all superfluous and ill-defined terms such as “family”, “groups/subgroups”, “types/subtypes”, and used only the terms “class” (BCs, Bistrat cells, AACs) and “subclass” (referring to the 6 subclasses listed in Figure 7) throughout the text. Regarding the number of AACs in the stratum oriens, please note that AACs are one of the rarest cell type in the adult hippocampus (7; 4), therefore, increasing the number of successfully recorded, visualized and immunocytochemically or electron microscopically identified AACs is an extremely difficult challenge. Nevertheless, in an effort to address the Reviewer’s concern, we carried out a new series of experiments and, after a lot of effort, managed to record two new AACs.

However, both of them were classical AACs situated in the pyramidal cell layer, not E-AACs. The ripple-related firing of these two new cells are now included. Regarding the dendrites of the AACs, there were no obvious differences; both C-AACs and E-AACs had dendrites from the stratum lacunosum-moleculare to the stratum oriens/alveus (Figure 5). Unfortunately, we cannot compare the total dendritic length of all the cells due to histological processing errors, and in vitro recordings are also problematic due to the truncation of the most distal dendrites.

*2) Although BCs with somata in the stratum oriens fire earlier than the other PV-cells and therefore might, as suggested by the authors provide perisomatic inhibition onto the other PV cells which finally may result in the observed 1.4 ms delay, the functional relevance of this delay in respect to encoding of information remains open. One explanation regarding the sub-classes in respect to PC innervation is that they may innervate different sub-classes of PCs (superficial versus deep PCs). This is an interesting hypothesis which the authors could have addressed in order to strengthen the functional significance of the various interneuron sub-classes. Again, this could be done under in vitro conditions*.

Please see our response to the related point #3 of Reviewer #1. In terms of functional relevance, a major finding in this paper is that BCs with somata outside of pyramidal layer (O-BCs) are capable of firing doublets of spikes during ripples, which is consistent with the suggested lack of effective perisomatic GABAergic inhibition from their classical counterparts (C-BCs). We have now included a sentence in the Discussion about potential functional relevance of basket cell doublets in synchrony (Traub et al., Nature 1996; the suggestion in the latter paper was that the doublet interval acts as a local circuit time constant that can match approximately the conduction delay between spatially separated circuits, and although the Traub paper focused on gamma, the mechanism, at least in principle, may be generalizable to faster, more local rhythms as well). With regards to the superficial versus deep PC question, we agree that this is a very interesting topic (Lee/Soltesz Neuron 2014) and we comment on the possibility that the various PV subclasses may be differentially involved in these subcircuits. However, a thorough experimental study of this issue is a major challenge and outside the scope of the present study.

*3) It is unclear and confusing that the upper gamma frequency range (90-130 Hz) is determined epsilon and not 'upper gamma'. The terminology 'upper gamma' is broadly used in the literature and corresponds in its relation to the theta phase exactly to the 'epsilon' activity of this study*.

We have included the rationale and explanation for our terminology in the manuscript. Briefly, regarding epsilon, we broadly followed the nomenclature in the recent publications by the Buzsaki lab (6; 72), where the running-associated fast gamma (>100Hz in rats) is also referred to as the epsilon band. This gamma(fast)/epsilon band has been shown to be largely local, arising from within the CA1 network, and a large number of CA1 pyramidal cells are phase-locked to the local gamma(fast)/epsilon oscillations (72).

Reviewer #3:

*One of the claims in the current manuscript is that AAC cells were not inhibited during ripples. In fact, none of the interneurons recorded in this or an earlier study (*[91]*) shows any ripple-inhibited neurons. Yet, ripple-inhibited interneurons are regularly observed in freely-moving rats, so this difference cannot be explained by anesthesia, which is one of the possibilities provided. For example,*
*Csicsvari et. al. 1999*
*(which should be discussed in this context) shows anti-SPW cells in 13 out of 81 cells in the alveus/oriens. The present and earlier studies features mainly cells from the pyramidal layer, but also some from oriens. Could it be due to minimal sampling of oriens cells in this account? A more detailed and reasoned explanation is necessary to help understand why there is such significant deviation from previously published reports*.

We don’t think that we selectively under-sample the oriens cells. In fact, as now pointed out, the population of PV interneurons in the stratum oriens has been estimated to be about 30% of the number of PV cells within the stratum pyramidale (in thousands: BCs: 1.3/3,9; Bistrat cells: 0.5/1.6; AACs: 0.3/1) (7), and the H-BC/C-BC and E-AAC/C-AAC ratios (3/10 and 2/5, respectively) in our sample approximated 30%, indicating that we did not under-sample the oriens layer cells. Second, there are specific interneurons, such as a subclass of dendritically projecting interneurons (Szabo, Varga and Soltesz, in preparation) that show uniform decreases in firing during ripples under our experimental conditions, and one of the C-AACs that we recorded for the revision of this manuscript also decreased its firing during ripples. Third, in the Royer et al. 2012 study, none of the optogenetically tagged PV cells decreased its firing during ripples (see supplementary Figure 8A in their study). Thus, we think that it is still unclear as to precisely what factor(s) may explain the fact that we observe generally fewer PV cells or OLM cells that decrease their firing during ripples compared to some, but not all, studies from other laboratories.

*Importantly, the use of 90-200 Hz for the “ripple” oscillation is not standard, as the authors acknowledge, and likely problematic. Sullivan et. al., which is cited to justify this frequency range, noted that 90-140 Hz fast gamma oscillations are distinct from sharp-wave ripples. For one, fast gamma oscillations seem to be local and low-amplitude, which some sharp-wave ripples resonate along the entire septo-temporal axis. The pooling of these events could account for the previously mentioned discrepancy with other studies-perhaps some cells (including Bistrat cells) do not fire during fast gamma, but only ripples. Judging from the examples in*
Figure 3*, the fast gamma (slow ripple) events appear to be of lower amplitude/power as well. This should be corrected to allow for proper evaluation and comparison of these results with other reports*.

We have now included the rationale and explanation for our terminology for the various fast oscillations in the manuscript. The key point is that there is evidence that the rest-associated, 90Hz-140Hz and 140Hz-200Hz fast oscillations share similar generation mechanism and differ mostly in the strength of incoming CA3 input (84), therefore, for the purposes of the current study we considered them together under the collective term ‘ripples’ (90Hz-200Hz). Importantly, this approach avoids the sorting of events according to a sharp, and likely at least to some extent artificial, 140Hz boundary into 90-140Hz or 140Hz-200Hz oscillations, and still allows the comparative study of the firing of interneurons during transient events that fall towards the lower- versus higher end of the continuum of 90Hz to 200Hz oscillations, as we have done in Figure 3. It is also important to point out here that we detected virtually identical number of events with filter settings of 90Hz-200Hz and 140Hz-200Hz, indicating that high gamma oscillations are leaking through classical ripple filters of 140Hz-200Hz.

*The authors did not perform any gamma oscillation analyses. The current account is incomplete as gamma is highly relevant for PV interneurons (see their 2012 PNAS study). For example, are the gamma phase preferences similar to epsilon phase preferences, or do they reflect similar temporal delays as in the earlier*
[91]
*study? It would also be interesting to know the gamma phase preferences for the AAC cells which do not fire during epsilon*.

Done. We have carried out an in-depth analysis of the PV interneuron discharges during gamma oscillations and also included the phase-preferential firing during gamma in the new summary figure (Figure 7).

[Editors' note: further revisions were requested prior to acceptance, as described below.]

*1) To support the lumping of the fast frequencies, please show a plot of the distribution of core frequencies for fast frequency events (> 90 Hz) to show that it is not a bimodal distribution composed of distinct fast gamma and ripple events (e.g. as reported by Sullivan et. al (*Figure 1*)*.

Done. The requested plot of the distribution of core frequencies is now shown in an inset in Figure 3, illustrating the lack of a bimodal distribution. A sentence has been added to the Results to describe the data: “Note that, as illustrated in the inset in Figure 3, rest-associated high-frequency events in our recordings showed no evidence of a bimodal distribution with a minimum at 140Hz, the frequency boundary used to separate high gamma and ripple events in sleeping or awake immobile rats (see Figure 1 in [84]).”

*2) Since Panel G of*
Figure 5
*has been significantly changed since the previous submission, please confirm that the xtic alignment is appropriate as it seems quite different from the previous version even considering the additional cells. It was noted that it appears that the ripple firing is in fact lower for these cells, particularly after the ripple onset. In Klausberger et. al.'s 2003 Nature paper the AAC cells were reported to fire biphasically during ripples. According to Klausberger et al, averaged across the ripple AAC cells “showed no significant difference in discharge frequency”, but they increased their firing at the onset of ripples, but were silent only after the ripple peak (see their*
Figure 4*). So it may be that the data does not differ from that reported by Klausberger et. al., as stated in the Discussion section. Please address this*.

Done. First, we thank the reviewer(s) for drawing attention to the fact that the xtic in Figure 5 was out of alignment, it is now corrected. Second, we have carried out an analysis of the AAC firing at the beginning of the ripples, and found that the AACs fired the majority of their ripple-related spikes during the first half of the ripples, which, as the Reviewer pointed out, is in agreement with [45]. We described the latter findings in a revised sentence: “Further analysis showed that the AACs fired the majority (68.3+-17.3%, n=7) of their ripple-related spikes during the beginning of the ripple events (before the ripple reached its maximal amplitude), in broad agreement with the discharge dynamics noted by Klausberger et al., 2003”. In addition, we further toned down disagreements with the Klausberger data in the Results and the Discussion.